# TOKEN-EFFICIENT ITEM REPRESENTATION VIA IMAGES FOR LLM RECOMMENDER SYSTEMS

**Kibum Kim**[1,✉]   **Sein Kim**[1,✉]   **HongSeok Kang**[1,✉]   **Jiwan Kim**[1,✉]
**Heewoong Noh**[1,✉]   **Yeonjun In**[1,✉]   **Kanghoon Yoon**[1,✉]   **Jinoh Oh**[2,✉]
**Julian McAuley**[3,✉]   **Chanyoung Park**[1,✉][*]

[1]KAIST   [2]Amazon Alexa   [3]UC San Diego

## ABSTRACT

Large Language Models (LLMs) have recently emerged as a powerful backbone for recommender systems. Existing LLM-based recommender systems take two different approaches for representing items in natural language, i.e., Attribute-based Representation and Description-based Representation. In this work, we aim to address the trade-off between efficiency and effectiveness that these two approaches encounter, when representing items consumed by users. Based on our observation that there is a significant information overlap between images and descriptions associated with items, we propose a novel method, **I**mage representation for **LLM**-based **Rec**ommender system (I-LLMRec). Our main idea is to leverage images as an alternative to lengthy textual descriptions for representing items, aiming at reducing token usage while preserving the rich semantic information of item descriptions. Through extensive experiments on real-world Amazon datasets, we demonstrate that I-LLMRec outperforms existing methods that leverage textual descriptions for representing items in both efficiency and effectiveness by leveraging images. Moreover, a further appeal of I-LLMRec is its ability to reduce sensitivity to noise in descriptions, leading to more robust recommendations. Our code is available at https://github.com/rlqja1107/torch-I-LLMRec.

## 1 INTRODUCTION

LLMs, which have recently shown remarkable performance in various NLP tasks by leveraging strong semantic reasoning and world knowledge, have inspired research into their application across diverse domains, including recommender systems (Kim et al., 2024a; Chen et al., 2024; Kim et al., 2024b; Xie et al., 2024). Especially, recent studies explore the replacement of traditional collaborative filtering models (e.g., SASRec (Kang & McAuley, 2018)) with LLMs as the backbone for recommendations (Lin et al., 2024b; Bao et al., 2023; Tan et al., 2024). To this end, they typically transform each item in a user's interaction history from a traditional numerical ID into natural language (e.g., titles) and arrange them into sequences within the input prompt.

The key to LLM-based recommender systems lies in effectively representing items in natural language to capture user preferences based on LLM's comprehensive understanding of user interaction history. In this regard, existing studies that represent items in natural language can be categorized into the following two main approaches: **1) Attribute-Based Representation** approach is to combine simple attributes such as brand and category with the item title to represent the item (Bao et al., 2023; Li et al., 2023a; Tan et al., 2024). For example, TALLRec (Bao et al., 2023) and TransRec (Lin et al., 2024b) enhance the understanding of user preferences by adding attributes, thereby facilitating a multifaceted understanding of items **2) Description-Based Representation** approach is to use detailed item descriptions[1] to preserve rich textual item semantics so that the LLM could capture user preference in a more fine-grained manner. For example, TRSR (Zheng et al., 2024b) summarizes full descriptions and provides them to the LLM for recommendation, thereby overcoming the limited item semantics provided by attributes only and addressing the input length constraints of

---

[*]Corresponding Author
[1]While attributes are the high-level, general features in a few keywords (e.g., Apple), descriptions generally provide item-specific details (e.g., It is a slim metallic body, 13-inch Liquid Retina, and black keyboard...).

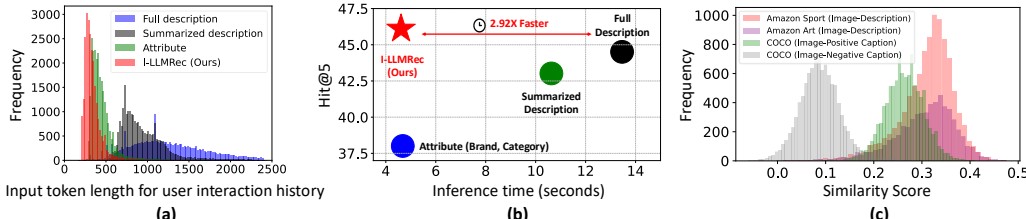

Figure 1: (a) Histogram of the input token length required to represent a user's item interaction history across different item expression approaches. (b) Recommendation performance (Hit@5) and Inference Time (seconds/100 users) for different item representation approaches. For (a) and (b), we use the Amazon Sports dataset for analysis. (c) Cosine similarity between item image-description pairs in Amazon Sport and Art datasets and image-caption pairs in the COCO dataset using CLIP.

LLMs. Overall, these approaches demonstrate the potential of natural language in enriching item representations for LLM-based recommendation.

However, *we argue that when representing items in natural language, there is an inherent trade-off between efficiency and effectiveness.* Specifically, the Attribute-Based Representation approach is efficient since it requires fewer tokens to add item attributes than to add the entire item descriptions, which shortens the input length for the LLM. However, it sacrifices the effectiveness of understanding user preferences due to the limited item semantics that can be contained in relatively short attributes. On the other hand, the Description-Based Representation approach is less efficient due to the longer input length required for detailed item descriptions, whereas the effectiveness of understanding user preferences is improved by incorporating richer textual semantics of items. In fact, as shown in Figure 1(a), the input length required for LLMs to represent a user's item interaction history significantly varies depending on how the item is represented. That is, the Description-Based Representation approach (i.e., Summarized description[2] and Full description) demands much longer input token lengths than the Attribute-Based Representation approach (i.e., Attribute), resulting in the increased computational complexity.

To better understand this trade-off, in Figure 1(b), we extend our analysis beyond a simple comparison of input token length and examine the computational complexity of different item representation approaches (i.e., efficiency) along with their recommendation performance (i.e., effectiveness). Note that we use the same LLM backbone (i.e., Sheared LLaMA-2.7B (Xia et al., 2023)) and adopt the same recommendation protocol for fair comparisons (See Appendix B regarding the details of the recommendation protocol). We observe that the Attribute-based Representation approach (blue circle) achieves the inference time more than 2.5 times faster than the Description-based Representation approach (green and black). However, this comes at the expense of a performance reduction of over 13% due to the lack of rich item semantics contained in attributes compared with descriptions. Furthermore, summarizing full descriptions reduces inference time but compromises performance due to the information loss during the summarization. In conclusion, the trade-off can be summarized as: *richer item representation provided to the LLM improves effectiveness but decreases efficiency*. However, this trade-off is unavoidable by nature if items are represented with natural language.

In this paper, we focus on addressing this trade-off based on our observation that there is a significant information overlap between images and descriptions associated with items. Specifically, when we measure the similarity between item image-description pairs in a real-world e-commerce dataset (i.e., Amazon Sport and Art (Ni et al., 2019)) using a vision-language model (i.e., CLIP (Radford et al., 2021)) in which image and language spaces are jointly embedded, we found that the similarity is surprisingly high. Specifically, Figure 1(c) shows that the average similarity for the Amazon Sport and Art datasets is around 0.31. To better understand this similarity level, we conducted the same experiment with the COCO caption dataset (Lin et al., 2014), a well-curated image-caption pair dataset widely used in vision-language research. Specifically, we measured similarities for both positive (i.e., image-caption) and negative (i.e., image-randomly sampled unrelated caption) pairs. Given that the negative pairs have an average similarity of only 0.07, the positive pairs' average similarity of around 0.26 can be considered an indicator of high similarity. Furthermore, the similarity

---

[2]To summarize the full description, we adopt an approach similar to TRSR (Zheng et al., 2024b) and use a similar prompt. For more details, please refer to Appendix A.

value of 0.31 between item image-description pairs, which is even higher, highlights a significant information overlap between images and their associated descriptions[3].

Building on this observation, we propose **I**mage representation for **LLM**-based **Rec**ommender system (I-LLMRec), which addresses both efficiency and effectiveness of existing LLM-based recommender systems by leveraging images as an alternative to lengthy textual descriptions for the item representation, aiming at reducing token usage while preserving the rich semantic information of item descriptions[4]. The main technical challenge lies in the misalignment between the item image space and the language space. To this end, we adopt a learnable adaptor for visual features followed by a technique to bridge the gap between the two spaces (i.e., Recommendation-oriented Image-LLM Semantic Alignment (RISA) module), which facilitates the training of the adaptor with carefully crafted prompts tailored to the recommendation context. This ensures the LLM to capture rich item semantics through images with only a few tokens, thereby effectively and efficiently capturing user preferences.

Through extensive experiments on real-world Amazon datasets, we demonstrate that our proposed method using images instead of item descriptions is superior in both effectiveness and efficiency. Specifically, I-LLMRec improves inference speed by approximately 2.93 times compared to the Description-based Representation approach, while achieving a 22% performance improvement over the Attribute-based Representation approach (Figure 1(b)). As a further appeal, I-LLMRec facilitates robust recommendations by mitigating sensitivity to noise associated with item descriptions as it leverages images. Our main contributions can be summarized as follows:

- We identify a trade-off between efficiency and effectiveness when expressing items in natural language.
- Based on the observation that there is a significant overlap between item image-description pairs, we propose a novel method, called I-LLMRec, which utilizes images instead of textual descriptions to efficiently and effectively capture user preferences.
- Our extensive experiments demonstrate that I-LLMRec outperforms the various natural language-based representation approaches in both effectiveness and efficiency.

## 2 PRELIMINARIES

Here, we introduce the task formulation and examine the complexity of different item expression approaches.

**Task Formulation.** Throughout this paper, we mainly focus on sequential recommendation, as it closely aligns with real-world scenarios (Tian et al., 2022; Xie et al., 2022). Let $\mathcal{U}$ and $\mathcal{I}$ denote the set of users and items, respectively. A user $u \in \mathcal{U}$ has the historical item interaction sequence denoted as $\mathcal{S}_u=[i_1, i_2, ..., i_k, ..., i_{|\mathcal{S}_u|}]$, where $i_k$ denotes the $k$-th interacted item and $|\mathcal{S}_u|$ is the number of items in the interaction sequence. The goal of this task is, for each user $u$, to predict the next item $i_{|\mathcal{S}_u|+1}$ to be consumed by the user based on the user interaction history $\mathcal{S}_u$.

**Representing Item Semantics.** With the advancement of LLMs for recommender systems, there is a growing need to help them understand the semantics of items to capture user preferences. In this regard, we categorize the multiple types of information to recommend an item $i$'s semantics into $[\mathbf{I}_i, \mathbf{D}_i, \mathbf{A}_i]$, where $\mathbf{I}_i$, $\mathbf{D}_i$, and $\mathbf{A}_i$ are an item image, textual description[5], and attribute, respectively.

**Comparison of Complexity.** Let $f(\cdot)$ denote the transformation function for the LLM to convert the representation of item semantics into input tokens, and let $|f(\cdot)|$ denote the number of input tokens. Specifically, for $\mathbf{D}_i$ and $\mathbf{A}_i$ that are in natural language, $f$ refers to tokenizing the natural language and converting them into the word embeddings. Meanwhile, for an image $\mathbf{I}_i$, $f$ involves extracting the visual features using a pretrained vision encoder (e.g., CLIP-ViT (Radford et al., 2021)), followed by an adaptor to resize their feature dimensions for compatibility with the LLM (Liu et al., 2024a; Li et al., 2023b). The complexity of LLM operations for each representation

---

[3]Such an observation was consistently observed across other real-world datasets as well as another CLIP variant model (See Appendix C and E.4).

[4]Some text-specific information that is missing in an image may be lost, but we find that in Section 4.2, this information has surprisingly little impact on recommendations.

[5]To avoid confusion, we will henceforth refer to the description as summarized description instead of full description unless stated otherwise.

is $O((|f(\mathbf{I}_i)||\mathcal{S}_u|)^2 d)$, $O((|f(\mathbf{D}_i)||\mathcal{S}_u|)^2 d)$, and $O((|f(\mathbf{A}_i)||\mathcal{S}_u|)^2 d)$, where $|f(\mathbf{D}_i)| >> |f(\mathbf{A}_i)|$. This is derived based on the complexity of Transformer, i.e., $O(n^2 d)$, where $n$ is the input token length and $d$ is the feature dimension of the LLM. It reveals that the complexity of the description-based approach is not merely being incurred with a single item; rather, it increases with the user sequence length $|\mathcal{S}_u|$, and its complexity even grows quadratically, making this approach impractical. Therefore, in this work, we propose to lower the complexity of expressing an item by leveraging its associated image using only a single token (i.e., $|f(\mathbf{D}_i)| \gg |f(\mathbf{A}_i)| > |f(\mathbf{I}_i)| = 1$ )[6] while preserving the rich semantics of the item description. This is based on our observation that there is a significant information overlap between the item image-description pairs (Figure 1(c)).

## 3    PROPOSED METHOD: I-LLMREC

In this section, we provide a detailed explanation of I-LLMRec. After introducing the initial recommendation-related task prompt, we begin by encoding the images of items consumed by users, and mapping them to an LLM through a learnable adaptor (Section 3.1). However, since these mapped visual features are not inherently aligned with the language space, we propose the Recommendation-oriented Image-LLM Semantic Alignment (RISA) module that trains the adaptor with carefully crafted prompts tailored for the recommendation context, resulting in effectively bridging the alignment gap (Section 3.2). Meanwhile, we propose the REtrieval-based Recommendation via Image features (RERI) module, a training approach that enables the LLM to directly perform recommendations from the item corpus based on the visual features (Section 3.3). Finally, we describe the overall training objectives and inference for recommendation (Section 3.4). Figure 2 shows the overall framework of I-LLMRec.

### 3.1    MAPPING ITEM IMAGES TO AN LLM

In this section, our goal is to map items consumed by users to an LLM using only a few tokens, allowing us to capture user preference efficiently and effectively. Specifically, given the interaction history of user $u$, i.e., $[i_1, i_2, ..., i_{|\mathcal{S}_u|-1}]$, and a target item $i_{|\mathcal{S}_u|}$, we convert the interaction history into a sequence of item images $[\mathbf{I}_{i_1}, \mathbf{I}_{i_2}, ..., \mathbf{I}_{i_{|\mathcal{S}_u|-1}}] \in \mathbb{R}^{(|\mathcal{S}_u|-1) \times H \times W \times 3}$, where $H$ and $W$ are the image height and width, respectively. Then, we adopt a pretrained visual encoder $V$ of a vision-language model to extract the visual features of each item, defined as $v_i = V(\mathbf{I}_i) \in \mathbb{R}^{d_v}$. This process makes the sequence of visual features $[v_{i_1}, v_{i_2}, ..., v_{i_{|\mathcal{S}_u|-1}}] \in \mathbb{R}^{(|\mathcal{S}_u|-1) \times d_v}$, where $d_v$ is the dimension of the visual feature.

**Adaptor network for visual features.** However, since the visual feature dimension $d_v$ is different from that of the LLM feature $d$, and their spaces are inherently misaligned, we introduce an adaptor network $M : \mathbb{R}^{d_v} \to \mathbb{R}^d$ to make visual features compatible with the LLM's feature dimension, allowing $v_i$ to be interactive with word embeddings within the LLM's input layer. More precisely, we process each item's visual feature $v_i$ through the adaptor $M$, transforming it into a sequence of visual features that match the dimensionality of the LLM. Formally, the sequence of mapped visual features is defined as $\bar{\mathcal{S}}_u^{\mathbf{I}} = [\bar{v}_{i_1}, \bar{v}_{i_2}, ..., \bar{v}_{i_{|\mathcal{S}_u|-1}}]$, where $\bar{v}_i = M(v_i) \in \mathbb{R}^d$.

**Designing a prompt to represent a user's item interaction history.** To represent a user's item interaction history with visual features, we carefully design a prompt, enabling the LLM to comprehensively understand the item semantics. Specifically, for each item $i$ in a user's item interaction history, we represent it in the prompt as $\mathbf{P}_i = \{\text{Title} : \mathsf{ITEM\_TITLE}, \text{Visual Representation} : [\mathsf{VISUAL}]\}$, where $\mathbf{P}_i$ is the representation of item $i$, $\mathsf{ITEM\_TITLE}$ is the title of the item $i$, and $[\mathsf{VISUAL}]$ is the placeholder of item $i$'s visual feature $\bar{v}_i$. More precisely, before being forwarded to the LLM's transformer layer, $[\mathsf{VISUAL}]$ is replaced with $\bar{v}_i$, while the natural language in $\mathbf{P}_i$ is converted into word embeddings. Note that following prior studies (Bao et al., 2023; Li et al., 2023a), we use the item title instead of the item's numerical ID. By enumerating $\mathbf{P}_i$ over the item interaction history of user $u$ (i.e., $[\mathbf{P}_{i_1}, \mathbf{P}_{i_2}, ..., \mathbf{P}_{i_{|\mathcal{S}_u|-1}}]$), we represent the user interaction based on item images where each image is expressed using a single token, allowing the LLM to efficiently and effectively capture the user preference.

### 3.2    RECOMMENDATION-ORIENTED IMAGE–LLM SEMANTIC ALIGNMENT (RISA) MODULE

To make the LLM effectively capture user preferences from item images, it is crucial to align visual features with the language space in a meaningful way, requiring adequate supervision of the adaptor

---

[6]In average, $|f(\mathbf{D}_i)|$ and $|f(\mathbf{A}_i)|$ are approximately 160 and 10 tokens, respectively.

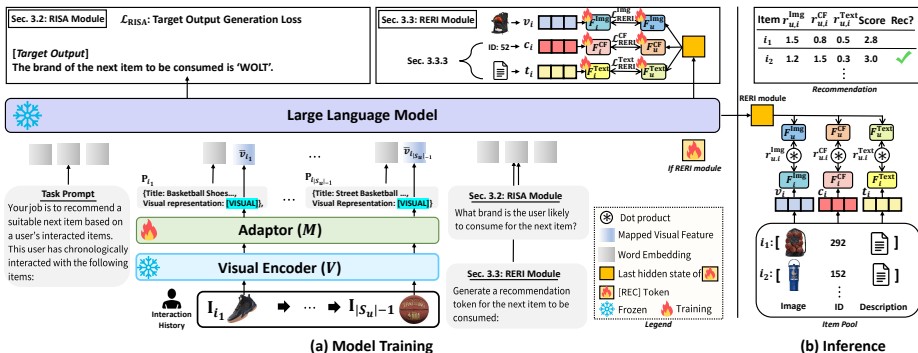

Figure 2: Overall framework of I-LLMRec. User-interacted item images are mapped into the LLM through an adaptor, which bridges the image and language spaces. To ensure alignment between two spaces, the adaptor is optimized via the RISA module. Furthermore, the recommendation process is formulated as a retrieval task via the RERI module.

network $M$. To achieve this, we optimize the adaptor network $M$ by predicting the next item properties based on the user interaction within a structured prompt, thereby guiding the LLMs to capture user preferences in the context of recommendation through the visual features.

Specifically, we craft a prompt in an *"input"-"target output"* format. Here, the *input* consists of the user interaction prompt, followed by a question regarding the next item (e.g., *What brand is this user likely to consume for the next item?*). The *target output* corresponds to the answer to that question (e.g., *The brand likely to be consumed is 'WOLT'*). To guide the LLM in considering various properties of the next item, we incorporate multiple properties of the next item, including brand, category, title, and description. For each property, we design five different question templates (refer to Appendix D.1 for complete templates). During each training step, we randomly select one of 20 possible question templates (4 properties × 5 templates) and train $M$ to generate the corresponding output. The training objective is formulated as $\mathcal{L}_{\text{RISA}} = \max_{M} \sum_{k=1}^{|y|} log(P_{\theta,M}(y_k|x, y_{<k}))$, where $\theta$ is the parameter of the LLM that is frozen, $y$ is the *target output*, $y_k$ is the $k$-th token of $y$, and $x$ is the *input*. Note that $x$ incorporates the image features of items consumed by users, which are supervised under $\mathcal{L}_{\text{RISA}}$. Please refer to Appendix D.2 for further discussion of LLM fine-tuning.

## 3.3 REtrieval-based Recommendation via Image features (RERI) module

We found that existing studies on LLM-based recommender systems often face challenges in providing *reliable* and *efficient* recommendations since they typically recommend items by predicting the next item title (Kim et al., 2024b; Lin et al., 2024b; Tan et al., 2024) or item tokens allocated as out-of-vocabulary (Wei et al., 2024; Li et al., 2023a; Zhu et al., 2024). Regarding *reliable* recommendation, the title prediction approach cannot guarantee that the recommended items exist within the item corpus, being likely to recommend non-existent items. Regarding *efficient* recommendation, title prediction relies on computationally intensive beam search to recommend multiple items, while the item token prediction approach requires an oversized LLM, as expanding the item pool necessitates extending the LLM vocabulary set, which hinders scalability. To overcome these challenges, we propose the REtrieval-based Recommendation via Image features (RERI) module, which formulates the recommendation task as a retrieval task by leveraging the images to directly retrieve relevant items from the item pool. In the following, we describe how to obtain an LLM-guided user representation for retrieval, and define a training objective that retrieves relevant items.

**LLM-guided user representation.** To obtain the LLM-guided user representation containing user preference, for each user $u$, we append an instruction prompt, i.e., *Generate a recommendation token for the next item to be consumed*, after the user interaction prompt $[\mathbf{P}_{i_1}, \mathbf{P}_{i_2}, ..., \mathbf{P}_{i_{|\mathcal{S}_u|-1}}]$. This instruction prompt helps generate a recommendation token, guiding the model towards item retrieval-based recommendations instead of item title generation-based recommendation. At the end of this prompt, we append a learnable token [REC] that aggregates a user's item interaction history and instruction-related information, leveraging the LLM's contextual reasoning capabilities. Specifically, we utilize the last hidden state associated with the [REC] token (i.e., $h([\text{REC}])$) to obtain the aggregated information of user preference for retrieval.

**Training objective for retrieving relevant items.** Given that $h([\text{REC}])$ represents the user's preference contained in the interaction history of user $u$, i.e., $[i_1, i_2, ..., i_{|\mathcal{S}_u|-1}]$, we formulate the task of retrieving the target item $i_{|\mathcal{S}_u|}$ based on a scoring function $\mathcal{T}$, unlike previous studies relying on a generative task (i.e., item title generation). To this end, we use the visual feature of items as the item representations and compute the affinity score between those and $h([\text{REC}])$. However, as $h([\text{REC}])$ and the visual features lie in the different spaces, we employ projectors for each, ensuring that they are mapped into a shared recommendation space. We denote the projectors and their outputs as $o_u^{\text{Img}} = F_u^{\text{Img}}(h([\text{REC}])), o_i^{\text{Img}} = F_i^{\text{Img}}(v_{i_{|\mathcal{S}_u|}})$, where $F_u^{\text{Img}} : \mathbb{R}^d \to \mathbb{R}^{d_s}, F_i^{\text{Img}} : \mathbb{R}^{d_v} \to \mathbb{R}^{d_s}$ are the projectors, and $d_s$ is the feature dimension of shared recommendation space. $o_u^{\text{Img}}$ and $o_i^{\text{Img}}$ denote the projected representations of user $u$ and item $i$ in terms of visual features, respectively. To optimize the retrieval task, we use the binary cross-entropy loss as follows:

$$\mathcal{L}_{\text{RERI}}^{\text{Img}} = -\sum_{u \in \mathcal{U}} \left[ log(\sigma(r_{u,i}^{\text{Img}})) + log(1 - \sigma(r_{u,i^-}^{\text{Img}})) \right] \tag{1}$$

where $r_{u,i}^{\text{Img}} = \mathcal{T}(o_u^{\text{Img}}, o_i^{\text{Img}}) = o_u^{\text{Img}} \circledast o_i^{\text{Img}} \in \mathbb{R}$ is the affinity score between $o_u^{\text{Img}}$ and $o_i^{\text{Img}}$, with $\circledast$ denoting the dot product, and $i^-$ is a randomly selected negative item. It is important to note that since we formulate the training objective as a retrieval task, we can guarantee that the recommended item exists in the item corpus and do not need to extend the item tokens within the LLM, resulting in a reliable and efficient recommendation process.

**Extension to multiple feature types.** Our retrieval-based recommendation approach can seamlessly incorporate multiple item feature types (e.g., ID-based embeddings and textual features) alongside visual features. Specifically, for a given item feature type denoted as $*$, we can simply integrate it by introducing two additional projectors, $F_u^*$ and $F_i^*$, such that $o_u^* = F_u^*(h([\text{REC}])), o_i^* = F_i^*(\textbf{IF}^*)$, where $\textbf{IF}^*$ represents the item feature corresponding to feature type $*$ (e.g., $v_{i_{|S_u|}} = \textbf{IF}^{\text{Img}}$ if $*$ is the image type $\text{Img}$), and $o_u^*$ and $o_i^*$ are the projected representations of user $u$ and item $i$ in terms of $*$ feature type, respectively. Using $o_u^*$ and $o_i^*$, we can then compute the binary cross-entropy loss $\mathcal{L}_{\text{RERI}}^*$ following Equation 1.

Among various item feature types, we extract the ID-based item embeddings from a pretrained collaborative filtering (CF) model (i.e., SASRec (Kang & McAuley, 2018)), thanks to its effectiveness in general recommendation tasks, especially for warm items (Kim et al., 2024b)[7]. Specifically, we incorporate ID-based item embeddings (i.e., $c_{i_{|S_u|}} = \textbf{IF}^{\text{CF}}$) in addition to the visual features (i.e., $v_{i_{|S_u|}} = \textbf{IF}^{\text{Img}}$). Furthermore, as item descriptions are readily available in real-world scenarios (Zheng et al., 2024b), we can easily incorporate textual features $t$ extracted from full descriptions to further improve overall performance (i.e., $t_{i_{|S_u|}} = \textbf{IF}^{\text{Text}}$). The analysis of extension to various features is provided in Section 4.4.

## 3.4 TRAINING AND INFERENCE

**Training.** With the image, CF, and text feature types, we combine the training objective from RISA module and three training objectives from RERI module, training the adaptor $M$ and projectors $F_u^*$ and $F_i^*$ ($* = [\text{Img}, \text{CF}, \text{Text}]$) while freezing the LLM. The combined loss[8] is denoted as:

$$\mathcal{L}_{final} = \mathcal{L}_{\text{RISA}} + \mathcal{L}_{\text{RERI}}^{\text{Img}} + \mathcal{L}_{\text{RERI}}^{\text{CF}} + \mathcal{L}_{\text{RERI}}^{\text{Text}} \tag{2}$$

**Inference.** For retrieval-based inference, we compute the affinity scores $r_{u,i}^{\text{Img}}$, $r_{u,i}^{\text{CF}}$, and $r_{u,i}^{\text{Text}}$ using the scoring function $\mathcal{T}$, in terms of visual, CF, and textual features, respectively. Specifically, the top-$k$ relevant items for $i_{|\mathcal{S}_u|+1}$ is computed as $rec_u^k = \text{Top-k}(r_{u,i}^{\text{Img}} + r_{u,i}^{\text{CF}} + r_{u,i}^{\text{Text}}), \forall i \in \mathcal{I}$, where $rec_u^k$ is the set of recommended $k$ items for user $u$, and Top-k is a function that extracts items with the highest top-$k$ scores. Note that while the affinity scores across different features can be aggregated through various approaches, such as weight summation or product, we opt for a simpler summation.

## 4 EXPERIMENTS

In this section, we conduct experiments to explore the following research questions:

---

[7]In Section 4.4, we show that the utilization of ID-based item embeddings leads to more recommendations of warm items.

[8]We opt to fix all weights at 1 to avoid the computational burden of tuning over a large hyperparamter space.

Table 1: Performance Comparison. **A**: Attributed-based Representation, **CF**: CF-based Representation (i.e., the CF item embedding is projected into the LLM), **D**: Description-based Representation, **I**: Image-based Representation.

| Dataset | Metric | Collaborative Filtering | | | | LLM | | | Image-aware LLM | | |
|---|---|---|---|---|---|---|---|---|---|---|---|
| | | GRU4Rec | VBPR | BERT4Rec | SASRec | TALLRec (A) | A-LLMRec (CF) | TRSR (D) | UniMP (I) | I-LLMRec (I)+D | I-LLMRec (I) |
| Sport | NDCG@5 | 0.2106 | 0.2369 | 0.2389 | 0.3129 | 0.2938 | 0.3352 | 0.3375 | 0.3364 | 0.3637 | **0.3711** |
| | Hit@5 | 0.2820 | 0.3097 | 0.2993 | 0.3841 | 0.3801 | 0.4070 | 0.4302 | 0.4030 | 0.4554 | **0.4570** |
| | NDCG@10 | 0.2413 | 0.2694 | 0.2670 | 0.3514 | 0.3323 | 0.3683 | 0.3765 | 0.3629 | 0.4003 | **0.4071** |
| | Hit@10 | 0.3775 | 0.4108 | 0.3866 | 0.4760 | 0.4997 | 0.5096 | 0.5515 | 0.4853 | **0.5694** | 0.5689 |
| Grocery | NDCG@5 | 0.2673 | 0.2321 | 0.2995 | 0.3753 | 0.3477 | 0.3860 | 0.3802 | 0.3710 | 0.3908 | **0.3956** |
| | Hit@5 | 0.3609 | 0.3171 | 0.3834 | 0.4684 | 0.4589 | 0.4823 | 0.4917 | 0.4506 | 0.5037 | **0.5069** |
| | NDCG@10 | 0.3033 | 0.263 | 0.3317 | 0.4096 | 0.3874 | 0.4195 | 0.4184 | 0.4011 | 0.4288 | **0.4332** |
| | Hit@10 | 0.4726 | 0.4232 | 0.4835 | 0.5746 | 0.5815 | 0.5864 | 0.6099 | 0.5439 | 0.6211 | **0.6232** |
| Art | NDCG@5 | 0.3119 | 0.3710 | 0.3626 | 0.4561 | 0.4572 | 0.4652 | 0.4758 | 0.4565 | 0.4796 | **0.4839** |
| | Hit@5 | 0.4044 | 0.4656 | 0.4455 | 0.5374 | 0.5663 | 0.5681 | 0.5841 | 0.5315 | **0.5902** | 0.5883 |
| | NDCG@10 | 0.3481 | 0.4062 | 0.3955 | 0.4860 | 0.4944 | 0.4981 | 0.5100 | 0.4829 | 0.5160 | **0.5191** |
| | Hit@10 | 0.5203 | 0.5745 | 0.5477 | 0.6301 | 0.6813 | 0.6877 | 0.6896 | 0.6131 | **0.6981** | 0.6974 |
| Phone | NDCG@5 | 0.2023 | 0.1898 | 0.2319 | 0.3299 | 0.3721 | 0.3403 | 0.3886 | 0.3388 | 0.3892 | **0.3900** |
| | Hit@5 | 0.2877 | 0.2688 | 0.3184 | 0.4366 | 0.4986 | 0.4502 | 0.5148 | 0.4427 | **0.5176** | 0.5156 |
| | NDCG@10 | 0.2353 | 0.2222 | 0.2664 | 0.3658 | 0.4148 | 0.3811 | 0.4292 | 0.3757 | 0.4309 | **0.4320** |
| | Hit@10 | 0.3952 | 0.3698 | 0.4255 | 0.5478 | 0.6305 | 0.5761 | 0.6401 | 0.5569 | **0.6463** | 0.6448 |

- **RQ1:** How does I-LLMRec perform compared to the CF and LLM-based recommender models?

- **RQ2:** How well does I-LLMRec offer strengths (i.e., efficiency, effectiveness, and robustness) by utilizing images rather than lengthy descriptions?

- **RQ3:** How does each module (i.e., RISA and RERI) contribute to the I-LLMRec?

- **RQ4:** How can we handle when item images are missing?

## 4.1 EXPERIMENTAL SETTING

**Datasets.** For evaluation, we use four categories from the Amazon dataset (Ni et al., 2019): Sports, Grocery, Art, and Phone. These datasets contain the item titles, attributes (e.g., brand and category), detailed textual descriptions, and images representing items. Following a prior study (Wei et al., 2024), we filter out users and items with fewer than 5 interactions to ensure data quality. We summarize the data statistics in the Appendix E.1.

**Evaluation Protocol.** Following the leave-one-out protocol (Kim et al., 2023; Kang & McAuley, 2018), we use each user's most recent interaction as the test set, the second most recent interaction as the validation set, and the remaining interactions as the training set. For evaluation metrics, we utilize Hit Ratio (Hit@$k$) and Normalized Discounted Cumulative Gain (NDCG@$k$) with $k = 5, 10$. Hit@$k$ measures whether the ground-truth item appears in the recommended list (i.e., $rec_u^k$) while NDCG@$k$ evaluates its ranking position. To reduce computational complexity during evaluation for each user, we randomly select 100 negative items that the user has not interacted with, add the ground-truth test item, and compute the metrics based on this set.

Regarding the Implementation Details and Baselines, please refer to Appendix E.2 and E.3.

## 4.2 PERFORMANCE COMPARISON (RQ1)

Table 1 shows the performance comparison between I-LLMRec and baselines across four datasets. The key observations are as follows: **1)** LLM-based models generally outperform traditional CF models, highlighting the effectiveness of leveraging an LLM backbone for the recommender systems. **2)** A-LLMRec and TRSR generally outperform TALLRec. It suggests that beyond the attributes, incorporating CF item embeddings or item descriptions further enhances the LLM's ability to capture user preferences. However, we observe that A-LLMRec performs worse on the Phone dataset, and TRSR generally outperforms A-LLMRec, indicating that item descriptions are more effective than ID-based item embeddings in expressing item semantics. **3)** I-LLMRec outperforms UniMP, an image-aware LLM-based model. Given that UniMP employs a multimodal LLM where images and texts are pre-aligned, it indicates the need for a more specialized alignment strategy tailored to the recommendation tasks, demonstrating the effectiveness of I-LLMRec. **4)** Considering that TRSR includes attributes during summarization (See Appendix A), adding description information in TALLRec, which is equivalent to TRSR, improved performance, whereas descriptions in addition to images (i.e., I-LLMRec+D[9]) did not improve the performance of I-LLMRec. This suggests that while descriptions convey useful information, they are inherently limited by what is already present in images due to the significant information overlap between image and description

---

[9]I-LLMRec+**D** is a variant of I-LLMRec that represent items using both image features and text description.

pairs. This supports the idea that item images are sufficient to capture user preferences for recommendation. For further experimental results on additional datasets, such as visual-centric datasets, please refer to Appendix E.4.

## 4.3 MODEL ANALYSIS (RQ2)

**Analysis on Efficiency.** To study the efficiency of I-LLMRec, we evaluate the inference time across different sizes of user's item interaction sequences (i.e., $|\mathcal{S}_u|$). Specifically, we divided the users into groups according to $|\mathcal{S}_u|$, randomly selected 100 users per group, and measured their total inference time. As shown in Figure 3, we observe that TRSR and I-LLMRec+D, which rely on lengthy descriptions, show a sharp

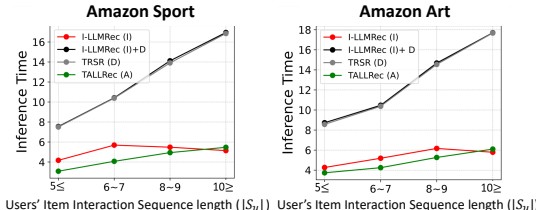

Figure 3: Inference time (sec/100 users) over $|\mathcal{S}_u|$.

increase in inference time as the length of textual descriptions within the prompt increases proportional to $|\mathcal{S}_u|$. On the other hand, I-LLMRec maintains consistently low inference time regardless of $|\mathcal{S}_u|$. This efficiency stems from the fact that even as $|\mathcal{S}_u|$ increases, only a minimal number of image tokens are added, keeping computational costs low. Furthermore, we observe that TALLRec, which is an attribute-based representation approach, shows a gradual increase in inference time and eventually surpasses I-LLMRec when $|\mathcal{S}_u| \geq 10$. We attribute this to the fact that processing natural language attributes scales in complexity faster than image processing required for I-LLMRec. In summary, I-LLMRec is more efficient than TRSR across all $\mathcal{S}_u$ and even more efficient than TALL-Rec when user interactions are high, demonstrating superior computational efficiency. Please note that this efficiency analysis includes the pre-computation of visual features, in line with real-world recommendation systems. For a detailed discussion on pre-computation, refer to the Appendix. E.5.

**Analysis on Effectiveness.** To gain a deeper understanding of the effectiveness of I-LLMRec, we examine how well it performs under a limited budget compared to the natural language-based approach. Specifically, we consider the LLM's context window size (i.e., maximum input token length) as the budget, and vary it from 4,096 to 256 tokens. This allows us to analyze the ability of models in effectively capturing the user preferences given a limited budget. As shown in Figure 4, we observe that as the context window size drops below 512,

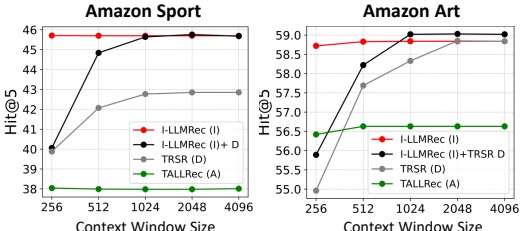

Figure 4: Performance of various LLM context window sizes.

the performance of TRSR and I-LLMRec+D declines sharply. This is because lengthy descriptions make it difficult for the LLM to fully process interactions with a limited context window size, thereby making it more challenging to understand user preferences. On the other hand, I-LLMRec maintains stable performance even at the context window size of 256 by representing item semantics using minimal image tokens. This enables the LLM to process full user interactions within the given budget and effectively capture user preferences. While TALLRec also remains stable at 256 due to its low token usage, it exhibits inferior performance because its attributes are not expressive enough. Overall, I-LLMRec effectively overcomes the context window size constraints by leveraging item images to preserve rich item semantics, demonstrating its effectiveness and practicality.

**Analysis on Robustness.** To evaluate the robustness of representing items using images instead of natural language as done in I-LLMRec, we compare its performance with various natural language-based approaches—TALLRec (Attribute), TRSR (Summarized description), and Full description—across multiple datasets. Figure 5 presents two key findings regarding the drawback of natural language-based approaches. **1) Information loss from summarization**

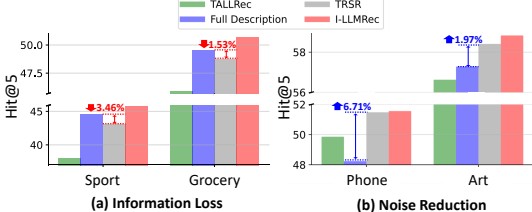

Figure 5: Performance of natural language-based approaches (TALLRec, TRSR and Full Description) and I-LLMRec on four datasets.

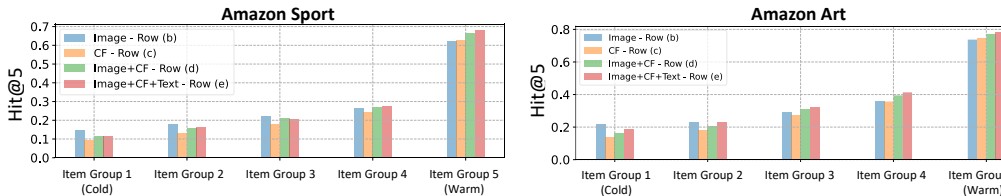

Figure 6: Performance across item groups, where higher IDs indicate warmer item groups.

(Figure 5(a)): Full description outperforms TRSR, indicating that summarizing descriptions introduces information loss, where essential semantic details may be omitted, leading to lower performance. **2) Presence of Noise in Full Description** (Figure 5(b)): On some datasets (i.e., Phone and Art), Full description rather performs worse than TRSR that relies on a summarized description. This indicates that the lengthy full description may contain noise and that the noise could be partially removed in the summarized version of the full description[10]. Such different behaviors of natural language-based approaches across different datasets demonstrates that these approaches are sensitive to the given text, and thus impractical in reality. In contrast, I-LLMRec, consistently delivers strong performance across all datasets, demonstrating its robustness in handling variations in item descriptions.

## 4.4 ABLATION STUDIES (RQ3)

In Table 2, we conduct ablation studies to understand the effectiveness of each component in I-LLMRec. The variant without any additional components is trained solely using Equation 1 and recommends items based only on image features (row (a)). We make the following observa-

Table 2: Ablation studies of I-LLMRec.

| Row | RISA | RERI | | | Sport | | Art | |
|---|---|---|---|---|---|---|---|---|
| | | Image | CF | Text | Hit@5 | NDCG@5 | Hit@5 | NDCG@5 |
| (a) | | ✓ | | | 0.3953 | 0.3043 | 0.5040 | 0.3915 |
| (b) | ✓ | ✓ | | | 0.4316 | 0.3403 | 0.5564 | 0.4447 |
| (c) | ✓ | | ✓ | | 0.4075 | 0.3256 | 0.5502 | 0.4517 |
| (d) | ✓ | ✓ | ✓ | | 0.4491 | 0.3630 | 0.5795 | 0.4769 |
| (e) | ✓ | ✓ | ✓ | ✓ | **0.4570** | **0.3711** | **0.5883** | **0.4839** |

tions: **1) Effect of RISA:** We observe that equipment with the RISA module significantly improves performance (row (a) vs. row (b)). This indicates that the RISA module effectively aligns visual features with the language space, enhancing the LLM's ability to understand user preferences. **2) Effect of extending to multiple feature types in RERI:** We observe that using two features together[11] yields better performance than using either the image or CF feature alone (row (b), (c) vs. (d)). Moreover, integrating all three feature types (i.e., Image, CF, and Text) further improves performance (row (d) vs. (e)). This demonstrates that extending multiple feature types enhances recommendation effectiveness beyond image features alone.

**Effectiveness of Feature types across cold/warm items.** To further analyze the impact of different feature types, we evaluate their performance on cold and warm item recommendations. Specifically, we follow (Liu et al., 2023b) by sorting the items in ascending order based on the frequency of interaction and dividing them into five equally sized groups, where a higher group ID indicates a warmer item group, Then, we compare the recommendation performance across these groups. As shown in Figure 6, we observe that using image features (blue bar) generally outperforms CF features (orange bar) for cold items (lower ID groups). However, as the item group transitions from cold to warm, CF's effectiveness improves, eventually surpassing Image in Item Group 5 (Warm). This suggests that CF is particularly effective for recommending warm items. With these insights, combining Image and CF (green bar) helps mitigate CF's weakness in recommending cold items, resulting in higher performance than CF alone in Item Group 1-2. At the same time, the CF features compensate for weakness of the image features in recommending warm items, leading to better performance than image alone in Item Group 4-5. Furthermore, adding textual features (red bar) further boosts performance across all Item Groups, exceeding the performance of Image+CF. This analysis highlights how different feature types complement each other to enhance recommendation.

## 5 DISCUSSION

**Rethinking Information Overlap.** We note that the observation of information overlap between image and text features, as discussed in Section 1, relates to the challenges of redundancy in mul-

---

[10]Indeed, we found that full descriptions often contain extraneous content, such as HTML tags introduced during data crawling.

[11]The inclusion of specific feature types is applied both in model training (Equation 2)

timodal recommender systems (Zhou et al., 2023; Yang et al., 2025; Jeong et al., 2024). Here, we discuss how this information overlap has been perceived in previous studies and articulate how our viewpoint departs from these interpretations. Specifically, prior studies on multimodal recommender systems generally view this information overlap as redundant information that is seen as a *barrier* to improving performance when both modalities are used. Therefore, they generally aim to extract complementary information from each modality to improve performance. On the other hand, we present a fundamentally different viewpoint, asserting that this information overlap is in fact a crucial factor in addressing the trade-off between efficiency and effectiveness when representing items for LLMs in natural language. In other words, we view this information overlap as an *advantage* for LLM-based recommender systems rather than a barrier. Moreover, this perspective allows us to maintain robust performance in real-world cases where item images are absent, as detailed in Appendix E.8 (**RQ4**).

## 6 RELATED WORK

**LLM-based Recommendation.** A myriad of studies have spurred the development of LLMs for recommender systems (Bao et al., 2023; Wang & Lim, 2023; Zhang et al., 2023; Liu et al., 2024b). Prior works adapt LLMs to recommendation by representing interaction history as item title sequences and enhancing performance with item attributes to enrich semantics. (**Attribute-based Representation**). Specifically, TALLRec (Bao et al., 2023) converts item IDs into their titles and captures user preference via LoRA (Hu et al., 2021) fine-tuning. TransRec (Lin et al., 2024b) improves the user preference understanding by including item titles and attributes in prompts. IDGen-Rec (Tan et al., 2024) incorporates item titles generated by an independent LLM for unique and meaningful semantics. More recently, to feed the richer item semantics to LLMs, a description of items have been utilized (**Description-based Representation**). Specifically, TRSR (Zheng et al., 2024b) utilizes a large LLM (i.e., LLaMA-30B-instruct (Touvron et al., 2023a)) to summarize item descriptions, and feeds them to a smaller LLM (i.e., LLaMA-2 7B (Touvron et al., 2023b)) to capture the condensed meaningful information, while ONCE (Liu et al., 2024b) adopts GPT-3.5 (Brown et al., 2020) for the summarizing task. Despite their notable success to capture the richer item semantics, they face a trade-off between efficiency and effectiveness when representing items in natural language, leading us to utilize image information to address this trade-off.

Please see Appendix F for additional related works (e.g., multimodal LLM-based recommendation).

## 7 CONCLUSION

In this work, we address the trade-off between efficiency and effectiveness when representing item semantics in natural language for an LLM-based recommender system. Based on the observation that there exists a significant information overlap between images and descriptions associated with items, we propose I-LLMRec, a novel method that leverages images to capture rich item semantics of lengthy descriptions with only a few tokens, thereby capturing user preferences efficiently and effectively. However, as the image and language space are not inherently aligned, we propose the Recommendation-oriented Image-LLM Semantic Alignment (RISA) module, which effectively bridges the gap between these spaces. Furthermore, we propose the REtrieval-based Recommendation via Image features (RERI) module, a retrieval-based recommendation approach, to enhance both the reliability and efficiency of the recommender systems. Our extensive experiments demonstrate that I-LLMRec outperforms natural language-based approaches in terms of both efficiency and effectiveness. Regarding the limitation and future work, please refer to Appendix G

### ACKNOWLEDGEMENT

This work was supported by Institute of Information & communications Technology Planning & Evaluation (IITP) grant funded by the Korea government(MSIT) (RS-2022-II220157, Robust, Fair, Extensible Data-Centric Continual Learning) as well as another IITP grant funded by the Korea govern- ment(MSIT) (RS-2022-II220077, Reasoning, and Inference from Heterogeneous Data).

## ETHICS STATEMENT

In accordance with the ICLR Code of Ethics, to the best of our knowledge, we have not encountered any ethical concerns in the course of this work. Furthermore, all datasets and pre-trained models used in our experiments are publicly available.

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

# Supplementary Material

### - Token-Efficient Item Representation via Images for LLM Recommender Systems -

> Given the item's metadata, including the title, attributes, and description,
> craft a concise summary:
> Title: {TITLE}
> Brand: {BRAND}
> Category: {CATEGORY}
> Description: {DESCRIPTION}
> Answer:

Figure 7: Prompt for summarizing descriptions. The red-colored text denotes the actual metadata for the item.

## A    PROMPT FOR SUMMARIZATION

Figure 7 shows the prompt used for description summarization. Following TRSR Zheng et al. (2024b), we incorporate attribute information along with the description to generate a concise summary.

## B    RECOMMENDATION PROTOCOL

In this section, we describe a comprehensive explanation of the recommendation protocol, which is consistently applied to Attribute-based Representation, Description-based Representation (i.e., Full Description and Summarized Description), and I-LLMRec as introduced in Section 1. Note that this recommendation protocol follows a retrieval-based recommendation approach similar to the RERI module discussed in Section 3.3. Therefore, we recommend reading Section 3.3 first for a deeper understanding. At a high level, we modify the way items are represented by replacing the visual representation with either an attribute-based or description-based representation within $\mathbf{P}_i$, followed by a recommending process similar to RERI module.

Specifically, we first craft the prompt, which includes the task prompt (as shown in Figure 2) and the semantics of user-interacted items in a sequence. Here, the item semantics depend on the item representation approach, which plays a crucial role in capturing user preferences. Then, we append a learnable token $[\text{REC}]$ at the end, allowing this token to aggregate the semantics of items the user has interacted with, thereby capturing user preferences. To obtain the corresponding representation, we use the last hidden state of $[\text{REC}]$ token, which serves as the LLM-guided user representation. Using this representation, we retrieve relevant items by comparing with items' visual, collaborative filtering (CF), and textual features. More precisely, for each feature type, we compute affinity scores by performing a dot-product between the LLM-guided user representation and item features. Based on the summation of affinity score across feature types for all items, we rank the items and recommend the Top-$k$ items with the highest scores.

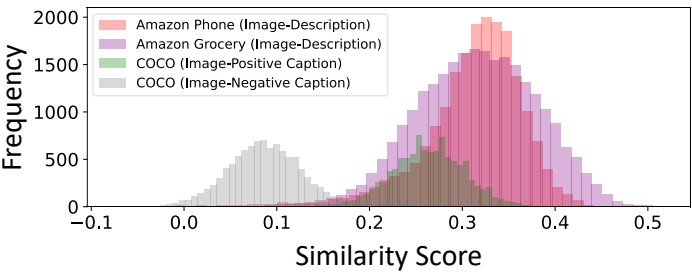

Figure 8: Cosine similarity between item image-description pairs in Amazon Phone and Grocery datasets and image-caption pairs in the COCO dataset using CLIP Radford et al. (2021).

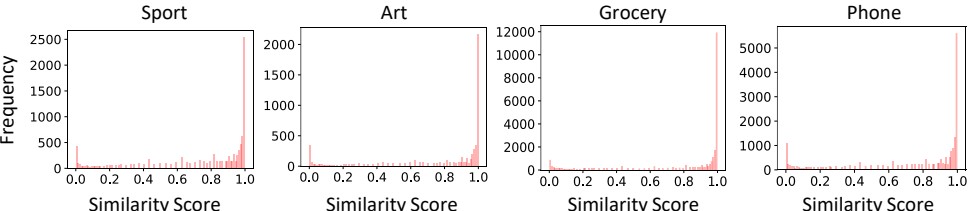

Figure 9: Sigmoid-based similarity (ranging from 0 to 1) between item image-description pairs across four datasets (Amazon Sport, Art, Grocery, and Phone datasets) using SigLIP Zhai et al. (2023).

## C CONSISTENT OBSERVATION OF INFORMATION OVERLAP

We further investigate the information overlap between image and description associated with items, which was explored in Section 1, by expanding these experiments to additional datasets and leveraging another CLIP-based model. **1) Expanding to additional datasets:** Building upon our previous comparative experiments in Amazon Sport and Art—where we used CLIP Radford et al. (2021) to measure cosine similarity in Section 1—we extend our analysis to the Amazon Grocery and Phone datasets. As shown in Figure 8, the average similarity for Amazon Phone and Grocery is approximately 0.31, which is higher than 0.26 average similarity observed in well-aligned COCO image-caption pairs. This consistent observation across multiple datasets further supports the information overlap between item images and descriptions. **2) Leveraging another CLIP model:** To further explore the information overlap using a different CLIP variant, we select SigLIP Zhai et al. (2023), a variant of CLIP with sigmoid loss function, to compute the similarity scores ranging from 0 to 1. Specifically, when we apply SigLIP to measure similarity between item image-description pairs across all four datasets (Amazon Sport, Art, Grocery, and Phone), we observed that, as show in Figure 9, the majority of items exhibit similarity scores close to 1, further demonstrating the significant information overlap between their image and description pairs.

## D PROPOSED METHOD: I-LLMREC

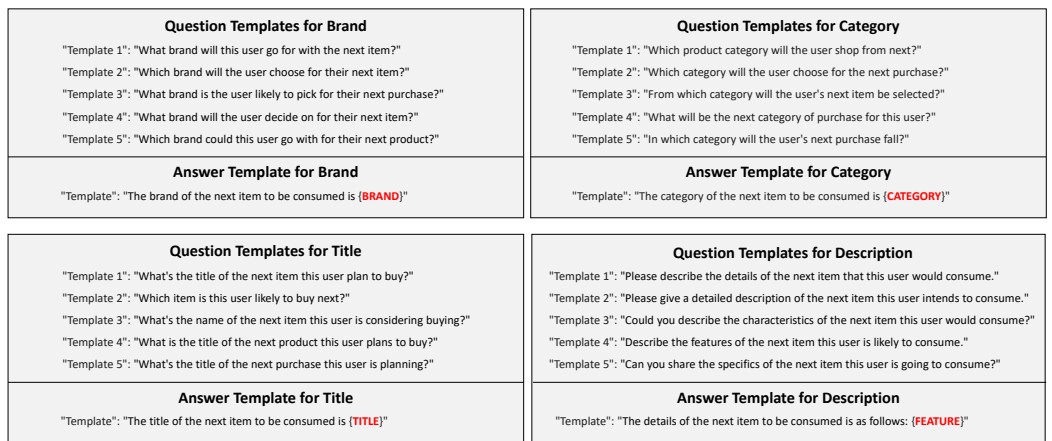

Figure 10: Prompt template for RISA module. The red-colored text denotes the actual information of an item.

### D.1 RISA PROMPT TEMPLATES

Figure 10 shows the prompt template for RISA module of I-LLMRec. As described in Section 3.2, we utilize the prompt templates to align the visual feature with the language space of LLMs by training the adaptor network $M$.

## D.2    DISCUSSION OF LLM-FINETUNING FOR I-LLMREC

A further benefit of I-LLMRec is that LLM fine-tuning is not mandatory, unlike the existing LLM-based recommender systems (Lin et al., 2024b; Bao et al., 2023; Tan et al., 2024; Zheng et al., 2024b; Liu et al., 2024b) that generally rely on costly fine-tuning of the LLM. Instead, we harness the LLM's intrinsic parametric knowledge to understand the user preferences. Throughout this paper, we opt to keep the LLM frozen to preserve the LLM's intrinsic knowledge and improve the training efficiency.

**Potential Improvement through LLM fine-tuning** Nevertheless, we explore potential performance gains from fine-tuning the LLM, a common approach in existing LLM-based recommender systems Lin et al. (2024b); Bao et al. (2023); Tan et al. (2024); Zheng et al. (2024b); Liu et al. (2024b). However, since fully fine-tuning an LLM demands substantial computational resources, we adopt the parameter-efficient fine-tuning strategy, LoRA Hu et al. (2021), which introduces a small number of learnable rank matrices into each LLM's transformer layer. As shown in Table 3, we observed the performance improvement through fine-tuning, indicating that there is room for further improvement within I-LLMRec. However, considering the computational cost, we opt to freeze the LLM.

Table 3: Performance comparison when LLM backbone of I-LLMRec is fine-tuned using LoRA Hu et al. (2021).

|  | Sport | | Art | |
|---|---|---|---|---|
|  | Hit@5 | NDCG@5 | Hit@5 | NDCG@5 |
| LLM Frozen | 0.4570 | 0.3711 | 0.5883 | 0.4839 |
| LLM fine-tuning with LoRA | **0.4633** | **0.3772** | **0.5958** | **0.4949** |

## D.3    DISCUSSION AND COMPARISON WITH SEMANTIC ID AND TEXTUAL IDENTIFIER APPROACHES

**Discussion with Semantic ID Approaches.**  Recently, generative recommender systems with semantic ID methods (Zheng et al., 2024a; Lin et al., 2025; Rajput et al., 2023; Kim et al., 2026; Hou et al., 2025) has been emerged. Here, we discuss the differences between our approach and semantic ID approach, and compare the efficiency and performance.

While both approaches share the goal of efficiently representing items within the LLM's capacity, there is a fundamental difference in how they convey the semantics of item descriptions to the LLM. The Semantic ID approach addresses the challenges of randomly initialized item tokens—which inherently fail to capture semantic relations between items (Liu et al., 2025)—by training a vector-quantized model (Zeghidour et al., 2021) that maps item descriptions to discrete codes. These descriptions serve as semantic intermediaries, aiding in the capture of semantic relationships between items (i.e., if two items share similar descriptions, they typically have the same codes), and allowing the LLM to *implicitly* capture the semantics of the descriptions. On the other hand, our approach *explicitly* encodes the semantics of each item's description using images, which provide a richer and more direct representation for the LLM.

Regarding token efficiency, Semantic ID approaches (Zheng et al., 2024a; Lin et al., 2025) generally require four tokens (codes) per item, whereas our approach uses a single image token that captures the semantics of the description. Importantly, Semantic ID approaches essentially require multiple discrete tokens to express cross-item semantic relationships, while our approach conveys the semantics of the description with just one image token. Beyond token efficiency, we also compare performance with a state-of-the-art semantic ID method, LC-Rec (Zheng et al., 2024a), under a fair evaluation setting using the same LLM backbone (Xia et al., 2023) on the Sport and Art datasets with full item candidates. I-LLMRec achieves Hit@5 scores of **16.28** on the Sport dataset and **21.80** on the Art dataset, compared to 15.86 and 20.41 for LC-Rec, respectively. These results indicate that I-LLMRec not only offers more efficient item representation but also more effective in representing item semantics to the LLMs.

**Discussion with Textual Item Identifier.** While Semantic ID approaches rely on latent codes (tokens) for item representation, the textual identifier methods (e.g., IDGenRec (Tan et al., 2024)) generally generate a paraphrased version of an item's title, derived from its textual description, to serve as its identifier. Similar to Semantic ID approaches discussed earlier, this approach also shares the goal of efficiently representing items. However, the textual identifiers inevitably lose the semantics of descriptions since they compress the full item description into a shortened title-like phrase. As a result, the state-of-the-art textual identifier method, IDGenRec, achieves only 15.11 Hit@5 on the Sport dataset and 17.97 on the Art dataset under the full item candidate setting—performance

that is inferior to I-LLMRec (16.28 and 21.80, respectively) and still below that of the semantic ID method LC-Rec.

Regarding token efficiency, IDGenRec compresses roughly 209 tokens of description into an average of 10.9 tokens per item. On the other hand, I-LLMRec uses a single visual token that directly encodes the item's semantic content. This leads to significantly lower computational cost during both training and inference, while still retaining richer semantic information.

## E EXPERIMENTS

Table 4: Data statistics after preprocessing.

| Dataset | # Users | # Items | # Interactions | Avg. Len |
|---------|---------|---------|----------------|----------|
| Sport | 24,665 | 9,884 | 179,491 | 7.28 |
| Grocery | 84,896 | 25,632 | 739,628 | 8.71 |
| Art | 14,986 | 5,801 | 121,917 | 8.13 |
| Phone | 59,307 | 19,671 | 403,194 | 6.80 |

### E.1 DATA STATISTICS

Table 4 shows the summary of the data statistics after preprocessing. Our experiments were conducted on datasets of varying scale, ranging from Grocery, which has a large number of interactions, to Art, which has relatively fewer interactions.

### E.2 IMPLEMENTATION DETAILS

We employed Sheared LLaMA 2.7B (Xia et al., 2023) as the LLM backbone. For the visual encoder $V$, we adopt SigLIP (Zhai et al., 2023), a variant of CLIP (Radford et al., 2021) with a sigmoid loss function, to extract visual features. Regarding experiments with different LLMs and visual encoders, please refer to Section E.12. We use a 2-layer MLP with ReLU activation function for the adaptor $M$ and the intermediate dimension is 512. In RERI module, we employ SBERT (Reimers, 2019) as the textual encoder. To ensure consistency across the LLM backbone, we use the same Sheared LLaMA 2.7B for TALLRec, A-LLMRec, and TRSR. Since these models are limited to recommending items only from a narrower item pool (i.e., 1 or 20), we apply the same RERI module for recommendation, which incorporates image, CF, and textual features, allowing them to recommend items from a larger pool. This approach is acceptable since our main goal is to evaluate how effectively the LLM captures user preferences through item expression. For TRSR, we use GPT-4o (OpenAI, 2024) to summarize full item descriptions. For UniMP, we follow its original implementation, which utilizes OpenFlamingo-4B-Instruct (Awadalla et al., 2023), a pre-trained multimodal LLM. Regarding training configurations, we set the learning rate to 0.001 and the batch size to 32. All models are trained on an NVIDIA GeForce A6000 48GB GPU.

### E.3 BASELINES

**Baselines.** We compare I-LLMRec with the following baselines:

*Collaborative Filtering (CF) models*.

- **GRU4Rec** (Hidasi, 2015) employs the RNN network to capture sequential user interactions.
- **VBPR** (He & McAuley, 2016b) enhances BPR (Rendle et al., 2012) by incorporating image features to improve personalized ranking.
- **BERT4Rec** (Sun et al., 2019) applies bidirectional transformers with masked item prediction to capture complex user preference.
- **SASRec** uses self-attention mechanisms to capture long-term user preference.

*LLM-based models*.

- **TALLRec** (Bao et al., 2023) converts numerical IDs into titles while intentionally adding attributes (i.e., brand and category), regarding it as a representative model of the Attribute-based Representation approach.

- **A-LLMRec** (Kim et al., 2024b) express items by projecting the CF item embeddings into the LLM along with title, enabling effective recommendation of warm items.

- **TRSR** (Zheng et al., 2024b) is a representative work of the Description-based Representation approach, which summarizes full item descriptions to effectively express items while leveraging rich textual semantics.

*Image-aware LLM-based models*.

- **UniMP** (Wei et al., 2024) utilizes image features and attributes to represent items, and recommends them through image tokens assigned as out-of-vocabulary.

- **I-LLMRec**+D (**equiv. to I-LLMRec+TRSR**)[12] is a variant of I-LLMRec that represent items using both image features and text description.

Note that the main difference between LLM-based models and I-LLMRec lies in how item semantics are represented (i.e., attributes, CF, descriptions, or images) in that both share the same LLM backbone and recommendation protocol.

Table 5: Performance results on three additional Amazon datasets (Automotive, Video, and Clothing) and the H&M Fashion dataset, with cosine similarity (Cosine Sim.) between item images and descriptions measured using CLIP (Radford et al., 2021) is measured.

| Model | Automotive (Cosine Sim.: 0.3086) | | Video (Cosine Sim.: 0.2801) | | Clothing (Cosine Sim.: 0.3107) | | H&M Fashion (Cosine Sim.: 0.2847) | |
|---|---|---|---|---|---|---|---|---|
| | Hit@5 | NDCG@5 | Hit@5 | NDCG@5 | Hit@5 | NDCG@5 | Hit@5 | NDCG@5 |
| SASRec | 0.4027 | 0.3078 | 0.6045 | 0.4713 | 0.4981 | 0.4690 | 0.4918 | 0.3885 |
| TALLRec | 0.4203 | 0.3107 | 0.6040 | 0.4521 | 0.4388 | 0.3660 | 0.3959 | 0.2827 |
| I-LLMRec | **0.4515** | **0.3379** | **0.6413** | **0.4915** | **0.5469** | **0.4927** | **0.5321** | **0.3949** |

### E.4 ANALYSIS ON OTHER DATASETS

**Behavior on visual-centric datasets.** We evaluate on two visual-centric datasets (Amazon Clothing and H&M Fashion datasets), where users' choices are likely influenced by visual attributes, and two less visual-centric datasets (Amazon Automotive and Video) to examine how I-LLMRec performs in visual-centric datasets compared to baselines, including the collaborative filtering model (SASRec(Kang & McAuley, 2018)) and natural language-based LLM approach (TALLRec (Bao et al., 2023)). As shown in Table 5, we observe that TALLRec performs on par with SASRec on the less visual-centric datasets, while showing a significant performance drop on visual-centric ones. It indicates that natural language alone is insufficient to represent items for LLMs in vision-driven contexts, making it challenging to capture user preferences driven by visual attributes. I-LLMRec, however, effectively leverages image information, yielding significantly higher performance than TALLRec, especially on visual-centric datasets. These results highlight the particular effectiveness of I-LLMRec in vision-driven contexts, while its solid performance on Automotive and Video further demonstrate its general effectiveness.

**Generalization of the observation.** To further demonstrate the information overlap between item images and descriptions discussed in Section 1, we measured the cosine similarity between images and descriptions on three additional Amazon datasets (Automotive, Video, and Clothing) as well as the H&M Fashion dataset (Ling et al., 2022) from another domain. As shown in Table 5, the high cosine similarity (Cosine Sim.) reported under each dataset consistently reveals a strong semantic overlap between images and descriptions, alongside superior performance of I-LLMRec over the baselines. While we could not cover every real-world dataset, the consistent results across eight datasets—including the four in the main paper—demonstrate the general applicability of information overlap, suggesting that I-LLMRec can be broadly applied.

---

[12]For each item in the user interaction, we append the description as "Description: $\mathbf{D}_i$" after $\mathbf{P}_i$ within the prompt, i.e., $\mathbf{P}_i = \{$Title : ITEM_TITLE, Visual Representation : [VISUAL], Description : $\mathbf{D}_i$ $\}$

**Generalization to the Video Domain.** We extended our evaluation to the micro-video recommendation domain by using the MicroLens (Ni et al., 2025) dataset and compared the performance with SASRec and TALLRec baselines. For visual features $v$, we average the image features of sampled frames provided by the MicroLens dataset. The results, as shown in the Table 6, show that TALLRec performs similarly to a traditional collaborative filtering model (SASRec), highlighting

Table 6: Performance on MicroLens dataset.

|          | Hit@5      | NDCG@5     |
|----------|------------|------------|
| SASRec   | 0.4567     | 0.3413     |
| TALLRec  | 0.4459     | 0.3237     |
| I-LLMRec | **0.4854** | **0.3520** |

that user preferences based solely on titles are insufficient. On the other hand, I-LLMRec significantly outperforms TALLRec, demonstrating the generalization of I-LLMRec to the video domain.

**Behavior on Visually-Weak Domains.** To explore the behavior of I-LLMRec on visually weak domains, we conducted additional experiments on the Goodbooks[13] dataset with multiple baselines (SASRec, TALLRec, and TRSR). As shown in Table 7, the description-based model TRSR outperforms I-LLMRec on this dataset. This aligns with intuition: in visually weak domains, visual features carry limited information for capturing user preferences, and textual descriptions naturally become more informative. However, we highlight two

Table 7: Performance on Goodbooks dataset.

|          | Hit@5      | NDCG@5     |
|----------|------------|------------|
| SASRec   | 0.4475     | 0.2956     |
| TALLRec  | 0.4752     | 0.3266     |
| TRSR     | **0.5100** | **0.3551** |
| I-LLMRec | 0.4870     | 0.3372     |

important points. First, while long textual descriptions can help in fully text-driven domains like Goodbooks, relying on them is often impractical due to significant computational cost. For example, TRSR incurs 2.8× longer inference time and 3.7× longer training time than I-LLMRec on Goodbooks. Furthermore, although we can leverage descriptions in such domains due to their low image–description cosine similarity (0.16 compared to over 0.29 in other datasets), these domains are limited in scope. Second, across the eight datasets used in the paper—including Sport and Phone, which can also be considered visually weak—we consistently observe the high degree of semantic overlap between item images and descriptions, and I-LLMRec performs effectively, demonstrating that I-LLMRec generalizes beyond purely visually rich domains (e.g., Fashion). In summary, the efficiency and effectiveness of I-LLMRec across diverse domains make it broadly more suitable for real-world recommendation scenarios.

### E.5   REGARDING THE PRECOMPUTATION OF VISUAL FEATURES

Here, we clarify how visual feature extraction is handled in our efficiency evaluation. Following prior multimodal recommendation studies (Xv et al., 2024; Xu et al., 2025; Lin et al., 2024a), we pre-compute and cache all item visual features before both training and inference. Thus, the wall-clock times in Figure 3 of the main paper exclude visual feature extraction cost. However, we argue that this setup aligns with real-world recommendation scenarios: once an item's visual features are extracted offline, the same representation can be reused for every user who interacts with that item, during both training and inference. Prior work (Xv et al., 2024; Xu et al., 2025) adopts the same convention when evaluating efficiency. Furthermore, it is important to note that caching these features is inexpensive—only 3.4KB per item, making the memory requirement extremely lightweight.

To conduct a thorough comparison of inference time with the description-based baseline (TRSR), we measure the inference time including visual feature extraction. On the Sports dataset, TRSR takes approximately 47 minutes for recommendation inference, while I-LLMRec requires 15 minutes for recommendation inference and an additional 5 minutes for visual feature extraction, totaling around 20 minutes. Even with this setup, I-LLMRec is 2.4 times more efficient.

### E.6   EFFECTIVENESS OF IMAGE FEATURES ON UNSEEN ITEMS

Extending the analysis of cold-start items in Section 4.4 of the main paper, to further demonstrate the effectiveness of image features for cold-start items, we evaluate the Sports and Art datasets by selecting users whose *test items do not appear in the training set* (i.e., extreme cold-start case), resulting in 302 users in Sports and 29 in Art. In this setting, using only CF features in the RERI module leads to near-zero performance, since unseen items retain untrained initial embeddings. On the other

---

[13]https://www.kaggle.com/datasets/zygmunt/goodbooks-10k

hand, image features achieve a Hit@5 of 0.3245 on Sports and 0.0689 on Art, demonstrating their effectiveness in recommending extreme cold start items.

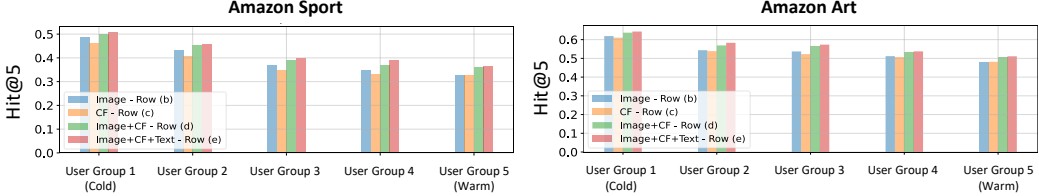

Figure 11: Performance across user groups, where higher IDs indicate warmer user groups.

### E.7 EFFECTIVENESS OF FEATURE TYPES ACROSS COLD/WARM USERS

We examine how different feature types affect performance under both cold- and warm-user recommendation settings. Following the same protocol as in Appendix 4.4 of the main paper, we partition users into groups based on their sequence length $|\mathcal{S}_u|$, ensuring that each group contains the same number of users. As shown in Figure 11, image features (blue bars) consistently outperform CF features across all user groups, indicating that image features remain effective regardless of the user's interaction history length. Moreover, incorporating more feature types—progressing from Image+CF (green bar) to Image+CF+Text (red bar)—yields steady performance gains in every group, demonstrating that leveraging the full set of features is broadly beneficial. Interestingly, we observe a gradual performance drop across all feature types as user sequence length increases, which contrasts with trends commonly observed in traditional CF models (Kim et al., 2023). We hypothesize that this decline arises from the difficulty LLMs face in capturing user preferences when longer histories introduce greater diversity in the categories of interacted items. Addressing this challenge—accurately modeling user preferences as interaction histories grow—remains an important direction for future work.

### E.8 MISSING IMAGE SCENARIO

In this section, we explore how to handle scenarios where item images are missing and evaluate how effectively we can manage them. Specifically, we randomly remove 50% of the item images from the entire item pool. As a result, these missing images cannot be used for item semantics rep-

Table 8: Performance when item images are missing.

|  | Sport | | Art | |
|---|---|---|---|---|
|  | Hit@5 | NDCG@5 | Hit@5 | NDCG@5 |
| TALLRec | 0.3801 | 0.2938 | 0.5663 | 0.4572 |
| TRSR | 0.4302 | 0.3375 | **0.5841** | 0.4758 |
| I-LLMRec w/ missing images | **0.4451** | **0.3589** | 0.5805 | **0.4789** |

resentation ($\mathbf{P}_i$) and retrieval-based recommendation (Equation 1 and $rec_u^k$ in the main paper). To compensate for the missing images, we substitute them with readily available textual descriptions. Similarly, for retrieval-based recommendations, when item images are missing, we extract textual features derived from descriptions using SigLIP (Zhai et al., 2023) text encoder rather than extracting visual features from images using SigLIP visual encoder. As shown in Table 8, I-LLMRec trained with half of the items missing images outperforms the baselines (TALLRec and TRSR), even though the baselines were trained without any missing item images. This demonstrate the effectiveness of I-LLMRec in handling the missing image scenario, which shows the practicality of I-LLMRec in reality.

### E.9 EXPLORING SENSITIVITY TO IMAGE QUALITY

Throughout the paper, we use high-resolution images, i.e., high-quality item images. However, in the real-world applications, it may not always be feasible to obtain such high-quality item images, and systems may instead rely on low-quality ones. To examine this issue, we investigate how I-LLMRec performs when the low-quality images are used. Specifically, we

Table 9: Performance under low-quality images.

| Row | Level of Image Quality | Hit@5 | NDCG@5 |
|---|---|---|---|
| | **Using only image features** | | |
| (a) | I-LLMRec w/ low quality images | 0.3704 | 0.2936 |
| (b) | I-LLMRec w/ high quality images | 0.4316 | 0.3403 |
| | **Using all features (Image, CF, Text)** | | |
| (c) | I-LLMRec w/ low quality images | 0.4459 | 0.3597 |
| (d) | I-LLMRec w/ high quality images | 0.4570 | 0.3711 |

replace high-resolution images with low-resolution versions originally provided in the metadata. We evaluate this setting on the Amazon Sport dataset under two conditions: (i) using only image features in both training and inference to isolate the effect of image quality, and (ii) using all features[14] (image, text, and CF). As shown in Table 9, we observe that performance drops significantly when relying solely on image features (row (a) vs. (b)), while the model remained robust when other features were included (row (c) vs. (d)). These observations indicate that while I-LLMRec is sensitive to image quality when used in isolation, it can still effectively handle low-quality images by leveraging complementary information.

### E.10 TRAINING STRATEGY

Throughout this paper, we optimize our model using an end-to-end training strategy to ensure ease of implementation and learning efficiency. However, in this section, we explore the impact of an alternative two-stage training strategy, where we separate the RISA and RERI modules. Specifically, for stage 1, we

Table 10: Training Strategy.

| Training Strategy | Sport | | Art | |
|---|---|---|---|---|
| | Hit@5 | NDCG@5 | Hit@5 | NDCG@5 |
| End-to-End | 0.4570 | 0.3711 | 0.5883 | 0.4839 |
| Stage1+Stage2 | 0.4604 | 0.3747 | 0.5858 | 0.4887 |

train the adaptor $M$ through the RISA module to initially align the visual space with the language space. For stage 2, we freeze the adaptor and train only $F_u^*$ and $F_i^*$ ($*$=[Img, CF, Text]) through the RERI module. As shown in Table 10, the two-stage strategy shows performance comparable to the end-to-end strategy, indicating that neither strategy offers a clear advantage over the other. Given this, we opt for the end-to-end training strategy to facilitate the ease of implementation and training efficiency.

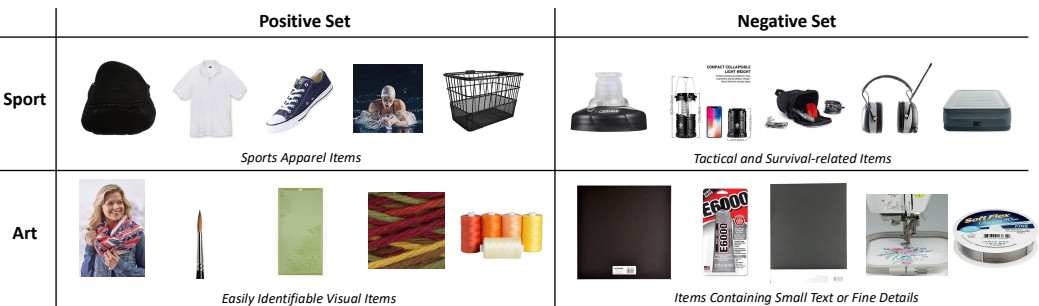

Figure 12: Case studies of impact of figure types in images.

### E.11 CASE STUDY ON THE IMPACT OF FIGURE TYPES

To study which figure types influence performance, we analyzed the item histories of users who correctly predicted the test item and selected their top 5 most frequently consumed items (positive set), as well as the top 5 items from users who failed to predict the test item correctly (negative set), using the Amazon Sport and Art datasets. We hypothesize that items in the positive set help LLMs capture user preferences, whereas items in the negative set have the opposite effect. As shown in Figure 12, we observe that in the Sport dataset, the positive set mainly consists of sports apparel (e.g., shoe, athletic shirts), while the negative set includes tactical or survival-related gear (e.g., camping pillow, lightbulb). This suggests that the functional category or intended use of an item appears to have an influence on performance. On the other hand, in the Art dataset, items in the positive set typically feature clear and easily recognizable visuals (e.g., paintbrush, yarn), whereas the negative set contains items that require interpreting small text or fine details to identify (e.g., a dark chipboard with a small label). These findings indicate that images providing clear visual cues are more beneficial for the model than those requiring detailed or text-based interpretation, instead of the functional category effect observed in the Sport dataset. In conclusion, the types of figures within images that affect performance seem to vary depending on the dataset.

---

[14]The LLM takes only the item images consumed by users as input, whereas the retrieval-based recommendation leverages image, text, and CF features.

### E.12 Impact of Different LLMs and Vision Encoders

To investigate the impact of the backbone models on the recommendation performance of I-LLMRec, we evaluated our model by replacing different backbone models. For the LLM, we used Sheared LLaMA-2.7B Xia et al. (2023) and Vicuna 1.5-7B Chiang et al. (2023),

Table 11: Impact of the different backbones.

| LLM | Vision Encoder | Sport | | Art | |
|---|---|---|---|---|---|
| | | Hit@5 | NDCG@5 | Hit@5 | NDCG@5 |
| LLaMA-2.7B | SigLIP | 0.4570 | 0.3711 | 0.5883 | 0.4839 |
| LLaMA-2.7B | CLIP | 0.4476 | 0.3560 | 0.5836 | 0.4796 |
| Vicuna 1.5-7B | SigLIP | 0.4670 | 0.3779 | 0.5943 | 0.4930 |
| Vicuna 1.5-7B | CLIP | 0.4552 | 0.3677 | 0.5885 | 0.4892 |

an improved version of LLaMA-7B Touvron et al. (2023a). For the vision encoder, we used CLIP[15] Radford et al. (2021) and SigLIP[16] Zhai et al. (2023), with the advanced version of CLIP scaling with sigmoid loss. Table 11 shows the following observations: 1) Given the same vision encoder, I-LLMRec with Vicuna 1.5-7B outperforms the model with LLaMA-2.7B. This improvement is likely attributed to Vicuna 1.5-7B's superior semantic reasoning capability, allowing it to capture user preferences more effectively. 2) Given the same LLM backbone, I-LLMRec with SigLIP performs better than the version with CLIP. This suggests that models that better capture visual features are more effective in delivering item semantics, thereby enabling the LLM to better understand the item semantics. In summary, employing more powerful LLMs and vision encoders leads to improvements in recommendation performance, as these models effectively capture item semantics and user preference.

### E.13 Simulation of Scaling Item Catalog

We explore how I-LLMRec manages the scalability of the item catalog by simulating its performance on the Amazon Phone dataset. In this simulation, we gradually increase the item catalog size from 5K to 20K (the full set). For a realistic setting, we assume that for each catalog size, there is a corresponding user base that has interacted solely with the items in that catalog. Specifically, we model the following user counts: 2.7K, 13.8K, 33.7K, and 59.3K users for catalog sizes of 5K, 10K, 15K, and 20K, respectively. This approach simulates the growing user base of expanding online services.

The performance of the models trained on each item catalog shows values of 0.468, 0.463, 0.4701, and 0.5106, indicating relatively stable performance even as the catalog size grows. However, we recognize that frequent model retraining results in increased memory and computational costs. To mitigate this, we tested using a model trained on a smaller catalog to infer for a larger catalog (e.g., a model trained on 5K items inferring for 10K, and a model trained on 10K items inferring for 15K). This approach avoids the need for periodic retraining as the catalog size grows and assess how robustly past models can perform as the item catalog grows. We observe that the performance remains robust with values of 0.370 (5K model to 10K catalog), 0.422 (10K model to 15K catalog), and 0.444 (15K model to 20K catalog), respectively, thanks to the I-LLMRec's cold-start item recommendation ability (See Appendix E.5). These results suggest that I-LLMRec can operate efficiently without frequent retraining, significantly reducing both computational complexity and training costs. Regarding system performance, I-LLMRec achieves a throughput of 20 users per second, including the indexing of visual features1, across all catalog sizes. This assumes the use of a fast retrieval model to first retrieve relevant items from the larger pool, with I-LLMRec performing re-ranking within the top 100 candidates. The GPU memory requirements are as follows: 6.7GB to load the model, and 21GB of GPU memory during inference with a batch size of 16. In terms of storage, the model weights require 14.3GB, and the visual feature cache requires 3.4KB per item. Thus, for every 5K increase in the catalog size, an additional 17MB of storage is needed. However, as the item catalog grows, the I-LLMRec model weights replace the previous model weights, meaning that no additional storage is required. For cost per request, since LLM-based recommendation models are primarily used for re-ranking [1], the processing cost remains constant regardless of catalog size, as indexing the visual features is negligible with a complexity of $O(1)$. The key scaling bottleneck is the periodic retraining necessary to maintain performance as the catalog expands. Using a single NVIDIA GeForce A6000 48GB GPU, an additional 3 hours of training time is required for every 5K increase in catalog size. However, we anticipate that utilizing multiple GPUs for offline training and fine-tuning models trained on previous catalogs will substantially reduce training time, compared to the current approach, which retrains the model from scratch with each catalog size increase.

---

[15]https://huggingface.co/openai/clip-vit-large-patch14-336
[16]https://huggingface.co/google/siglip-so400m-patch14-384

## F    ADDITIONAL RELATED WORKS

**Sequential Recommendation.** Our setup closely aligns with the sequential recommendation, where a user's item interaction history is listed as a sequence in chronological order, and the goal is to predict the user's next interaction item. Early studies process user sequences via the Markov chain for sequential recommendation (He & McAuley, 2016a; Mahmood & Ricci, 2007; Wang et al., 2015). With the advancement of deep learning, RNNs were used for sequence modeling to capture user preference (Hidasi, 2015; Li et al., 2017), while studies using CNN treat previous interacted items' embedding matrices as images, enabling the convolutional operation to consider user sequences (Tang & Wang, 2018; Yuan et al., 2019; Kim et al., 2016). Recently, with the emergence of Transformer (Vaswani, 2017), the self-attention mechanism has been applied to sequential recommendations (Kang & McAuley, 2018; Sun et al., 2019; Xu et al., 2021). More recently, the focus has shifted towards leveraging rich side information associated with items (e.g., images or text) in the sequential recommendation. Specifically, TempRec (Wu et al., 2022) encodes textual information of items to strengthen item embeddings, while MMMLP (Liang et al., 2023) fuses visual and textual features in the user sequence to capture fine-grained user preferences.

**Multimodal LLM-based Recommendation.** Recent studies (Zhou et al., 2025; Ye et al., 2025; Wei et al., 2024) have explored leveraging visual information in LLMs for recommendation tasks. Specifically, MSRBench (Zhou et al., 2025) investigates the use of off-the-shelf Multimodal LLMs (e.g., GPT-4V (Achiam et al., 2023)) for recommendation in various settings, while MLLM-MSR (Ye et al., 2025) and UniMP (Wei et al., 2024) fine-tune the Multimodal LLMs (Awadalla et al., 2023; Liu et al., 2023a) developed in computer vision to improve recommendation performance. Nonetheless, these studies mainly emphasize leveraging Multimodal LLMs to merely boost performance by feeding image features into LLMs, offering limited analysis of image–description overlap and lacking effective alignment strategies between visual and language spaces tailored to the recommendation context.

## G    LIMITATION AND FUTURE WORK

In this paper, we mainly exploit item images to capture user preferences effectively and efficiently for LLMs. A potential limitation of this approach is the possible underutilization of textual information. Specifically, even if item images are sufficient to capture user preferences by offering rich semantics, textual descriptions may still provide complementary information—such as subtle attributes or contextual nuances—that are not easily conveyed by images. However, extracting such information requires carefully disentangling text-specific cues from overlapping visual content and isolating only those parts of text that contribute meaningfully to recommendation tasks, which is a challenging and non-trivial process. Although the contribution of this information is estimated to be small according to our results in Section 4.2, combining it with image features could further enhance recommendation performance. We therefore identify this as a promising direction for future research.

