# OpenReview forum: "Token-Efficient Item Representation via Images for LLM Recommender Systems"
_ICLR.cc/2026/Conference — ICLR 2026 Poster_

### Official Review · Reviewer_11jm · 2025-10-30

**Soundness:** 3
**Presentation:** 3
**Contribution:** 3
**Rating:** 8
**Confidence:** 5

**Summary:**

The paper studies how to represent items for LLM-based sequential recommendation and argues there’s an inherent efficiency–effectiveness trade-off between attribute-only text and description-rich text. It proposes I-LLMRec, which replaces long descriptions with images, passing a single visual token per item into a frozen LLM via a learned adaptor (RISA) to align visual features to the language space, and a retrieval objective (RERI) that directly retrieve items from the item pool by leveraging the item images.

**Strengths:**

- The efficiency-effectiveness problem is well-motivated, with supporting analysis and the observation of image-description semantic overlap.
- Extensive experiments on four Amazon dataset demonstrating the effectiveness of the proposed method, with ablation study proving the contribution of each module.
- The method is less sensitive to noisy descriptions and handles 50% missing images by backing off to text features, still outperforming text-only baselines.

**Weaknesses:**

- The real-world deployment trade-offs need to be reconsidered, and how about the latency of visual encoder cost, adaptor and LLMs.
- The paper analyzes item coldness, but an explicit study of short user histories (cold-start users) is unexplored.
- How about the performance of baselines if we add image information?

**Questions:**

- What is the average total token count per item instance in practice (title + delimiters + [VISUAL]) across datasets?
- Could you explore how TRSR/TALLRec perform title-generation with beam search (their native setting) and compare to your retrieval at equal candidate sizes?
- How about the scalability to large e-commerce datasets? The authors should discuss the complexity of this method.

---

> ### Author Response · Authors · 2025-11-21
> **(1/2) Author Rebuttal**
>
> ### **W1(Q3): Latency Analysis**
>
> The latency of the visual encoder for processing per item is 0.03 seconds,  adaptor is nearly 0 seconds, LLM is about 0.07 seconds Although the visual encoder incurs some latency, this can be effectively mitigated using the same strategy as in prior multimodal recommendation research [1, 2], where item visual features are pre-extracted and stored in a memory bank, enabling O(1) lookup during inference. Regarding Q3, we believe I-LLMRec scales well to large e-commerce platforms. More specifically, each image feature can be extracted in approximately 0.03 seconds and cached. Furthermore, since our method does not require costly LLM fine-tuning, the overall training process completes within 14 hours on the Sports dataset, demonstrating strong training-time scalability. In Section 2 (Line 143-157), we also provide a complexity analysis, and Figure 3 presents an efficiency evaluation both showing that I-LLMRec is scalable than existing baselines. Finally, by adopting a retrieval-based inference pipeline rather than relying on LLM-based title generation (e.g., A-LLMRec), our method substantially reduces inference overhead and further improves system-level scalability.
>
> [1] Improving Multi-modal Recommender Systems by Denoising and Aligning Multi-modal Content and User Feedback. Xv et al., KDD’24
>
> [2] COHESION: Composite Graph Convolutional Network with Dual-Stage Fusion for Multimodal Recommendation. Xu et al., SIGIR’25
>
> ---
>
> ### **W2: Experiments for Cold-start Users**
>
> Extending the cold-start item analysis in Appendix E.5, we examine the performance of cold-start user groups across different feature types, with the corresponding results provided in Appendix E.6. In summary, we observe that image features remain effective compared to CF features regardless of the user’s interaction history length, including for cold-start users. Furthermore, incorporating additional feature types yields steady performance gains across all groups, demonstrating that leveraging the full set of features is broadly beneficial.
>
> Interestingly, we observe a gradual performance drop across all feature types as user sequence length increases, which contrasts with trends commonly observed in traditional CF models [1]. We hypothesize that this decline arises from the difficulty LLMs face in capturing user preferences when longer histories introduce greater diversity in the categories of interacted items. Addressing this challenge—accurately modeling user preferences as interaction histories grow—remains an important direction for future work.
> We sincerely appreciate the reviewer for being able to find an important challenge in LLM-based recommender systems.
>
> [1] MELT: Mutual Enhancement of Long-Tailed User and Item for Sequential Recommendation. Kim et al., SIGIR’23
>
> ---
> ### **W3: Baseline with Image Information**
>
> To explore how the performance of baselines is affected by the addition of image information, we integrated image data into the description-based baseline TRSR, represented as I-LLMRec+D in Table 1. We observed that adding descriptions to TALLRec (without descriptions) led to a significant performance improvement in TRSR. However, when image information was added to the TRSR (with descriptions), we observed only a modest performance gain (I-LLMRec+D). As discussed in Section 1, we attribute this to the significant overlap between description and image information, where the description alone already captures user preferences effectively, resulting in minimal impact from the additional image data. However, we argue that, as discussed in Section 4.3, the description-based baseline introduces significant complexity, which led us to leverage item images to capture user preferences for LLMs efficiently and effectively.
>
> ### **Q1: Average total token count per item**
>
> To address the reviewer’s question, we computed the average total token count, which includes the title, delimiters, and visual tokens, across all items in the datasets (Sports, Art, Phone, and Grocery). The average token counts are 24.5, 30.7, 45.2, and 27.3 for the Sports, Art, Phone, and Grocery datasets, respectively. We found that, in the Phone dataset, the inclusion of company names and long device names (e.g., LG Optimus L3 E400/E405 Unlocked GSM Phone with Android …) generally results in longer titles. On the other hand, in the Sports dataset, relatively shorter equipment names (e.g., BSI Taxi Single Ball Tote Bag) tend to make titles shorter.

---

> > ### Comment · Reviewer_11jm · 2025-11-21
> >
> > Thanks for your rebuttal and I've raised my score.

---

> ### Author Response · Authors · 2025-11-21
> **(2/2) Author Rebuttal**
>
> ### **Q2: Title Generation Setting for TRSR and TALLRec**
>
> To address the reviewer’s request for an exploration of TRSR and TALLRec's performance in title generation with beam search, we conducted experiments using the Sports and Art datasets. In these experiments, both models were fine-tuned with LoRA, applying next-title prediction loss (i.e., language autoregressive loss) instead of the retrieval loss described in Equation (1) of the main paper. During inference, we used beam search to generate multiple recommendations, applying a constrained decoding method [1] to select from a fixed set of 100 negative candidates.
>
> The results, shown in the table below, demonstrate that retrieval-based recommendations consistently outperform title prediction-based recommendations. This suggests that retrieval-based approaches are more effective because they align more closely with the original recommendation objective. Furthermore, we observed that the title prediction approach is less efficient than the retrieval-based method, as beam search necessitates repeated scoring and updating of candidates at each step. These findings emphasize that retrieval-based recommendations are not only more effective but also more computationally efficient than title prediction methods.
>
> [1] IDGenRec: LLM-RecSys Alignment with Textual ID Learning. Tan et al., SIGIR’24
>
> | Method             | Sports (HR@5 / NDCG@5) | Art (HR@5 / NDCG@5) | Inference Time (100 users) |
> |--------------------|-------------------------|-----------------------|-----------------------------|
> | TRSR-Title Prediction         | 0.3392 / 0.2291         | 0.4650 / 0.3303       | 247.3 seconds               |
> | TRSR-Retrieval     | **0.4302** / **0.3375**         | **0.5841** / **0.4758**       | 11.5 seconds                |
> | TALLRec-Title Prediction      | 0.3012 / 0.1897         | 0.4435 / 0.3069       | 120.2 seconds               |
> | TALLRec-Retrieval  | **0.3801** / **0.2938**         | **0.5663** / **0.4572**       | 4.5 seconds                 |
>
> - Inference time is computed by averaging the inference times of 100 users from both the Sports and Art datasets.

---

> ### Author Response · Authors · 2025-11-27
>
> We thank the revere for raising the score to 10 and for your insightful feedback, which has greatly contributed to improving the quality of our paper. Also, we sincerely appreciate the time you have dedicated to this process.

---

### Official Review · Reviewer_1MNz · 2025-10-31

**Soundness:** 3
**Presentation:** 3
**Contribution:** 2
**Rating:** 6
**Confidence:** 4

**Summary:**

The paper proposes an LLM-based recommender framework that replaces lengthy product descriptions with product images as the primary carrier of semantic information to reduce token usage and inference overhead while preserving recommendation quality. The method aligns visual features to the language model’s embedding space through a lightweight adapter trained with a recommendation-oriented prompting scheme, and then replaces text generation with a retrieval-style ranking head that scores real candidate items.

**Strengths:**

1. **Insightful reframing of the input modality under tight token budgets.** The paper clearly identifies the practical bottleneck of long textual inputs for LLM recommenders and argues—backed by empirical analyses of image–text redundancy—that images can serve as compact proxies for much of the descriptive semantics; this reframing is valuable because it opens a path to maintaining recommendation quality while substantially reducing context length, especially in settings where text is verbose, noisy, or unevenly curated.

2. **Well-scoped, efficiency-oriented architecture that avoids over-generation.** By aligning image features to the LLM and using a retrieval head rather than free-form generation, the approach ensures that outputs correspond to real catalog items, simplifies deployment, and naturally supports score-level fusion with collaborative-filtering and optional textual features; this design choice balances effectiveness with operational efficiency and makes the method easier to integrate into existing candidate-generation and re-ranking stacks.

3. **Comprehensive within-setting analyses that illuminate robustness and design trade-offs.** The experiments, while on public datasets, go beyond headline metrics to examine sensitivity to context window size, the impact of missing or degraded images, the benefits in cold-start regimes, and ablations that separate the roles of the alignment adapter and retrieval module; these studies help practitioners understand where the method is most reliable and how to combine it with other modalities without assuming that discarding text is universally optimal.

**Weaknesses:**

1. **Limited evidence for scalability to industrial-scale deployments.** Experiments rely on small, public datasets and do not convincingly demonstrate behavior under production constraints such as large candidate pools, high user/item cardinalities, tight latency/service-level targets, and strict token budgets; to strengthen external validity, the paper should report system-level metrics (e.g., throughput, memory footprint, time-to-rank, cost per request) alongside $Hit@K$/$NDCG@K$ under increasing catalog sizes and candidate set growth, include stress tests with realistic traffic patterns and cold-start skew, and, if proprietary data are unavailable, simulate scale (e.g., catalog expansion, multi-tenant load) to approximate industrial conditions.

2. **Relying on images while discarding available text is counterintuitive and under-justified.** Although the paper argues that product images capture much of the semantic content of descriptions, in practice text often provides complementary signals (attributes, specs, usage constraints, safety notes) and is essential in low-visual domains; a more convincing case would compare early/late fusion and gating strategies, include confidence-aware fallbacks when image quality is poor or missing, analyze error cases where text helps, and broaden coverage to text-centric categories (e.g., books, apps) to verify whether image-only inputs remain competitive when visual cues are intrinsically weak.

3. **Comparisons lack strong compressed-text or lightweight language baselines.** It is premature to conclude that long descriptions should be replaced rather than compressed without benchmarking against competitive text pathways that control for compute and token budget; recommended baselines include (i) learned summarization to fixed length with budget-aware prompts, (ii) small/distilled LMs or sentence encoders with adapter/prefix tuning, (iii) classical text features (TF–IDF/BM25) with a learned re-ranker, and (iv) category/attribute tokenization with compact encoders; for fairness, the paper should equalize total trainable parameters, training wall-clock, and inference budget across modalities and report $NDCG@K$/$Hit@K$ under the same token caps to verify whether images truly dominate when text is efficiently compressed.

4. **Methodological fairness is unclear given image-side tuning and a frozen LLM.** The current setup tunes an image adapter while leaving the LLM unfine-tuned, which may disadvantage purely textual baselines and conflate modality choice with where adaptation capacity is allocated; to ensure apples-to-apples comparisons, the authors should either (a) freeze both pathways symmetrically or (b) allow comparable adaptation on the text side (e.g., LoRA, prefix-tuning, lightweight adapters) under a matched parameter/cost budget, and further ablate the contribution of the visual adapter versus the retrieval head to isolate where gains originate, ideally reporting robustness under domain shift and varying rates of missing or low-quality images.

**Questions:**

see weaknesses.

---

> ### Author Response · Authors · 2025-11-21
> **(1/3) Author Rebuttal**
>
> ### **W1: Limited Evidence for Scalability to Industrial-scale Deployment**
>
> To address the reviewer's concern about scalability for industrial-scale deployments, we have simulated the scalability of our system by progressively increasing the item catalog size from 5K to 20K (full set) on the Phone dataset. In this simulation, we assume that for each item catalog, there is a corresponding user base that has interacted only with the items in that catalog. Specifically, we model the following user counts: 2.7K, 13.8K, 33.7K, and 59.3K users for catalog sizes of 5K, 10K, 15K, and 20K, respectively. This approach simulates the growing user base of expanding online services.
>
> The performance of the models trained on each item catalog shows values of 0.468, 0.463, 0.4701, and 0.5106, indicating relatively stable performance even as the catalog size grows. However, we recognize that frequent model retraining results in increased memory and computational costs. To mitigate this, we tested using a model trained on a smaller catalog to infer for a larger catalog (e.g., a model trained on 5K items inferring for 10K, and a model trained on 10K items inferring for 15K). This approach avoids the need for periodic retraining as the catalog size grows and assess how robustly past models can perform as the item catalog grows. We observe that the performance remains robust with values of 0.370 (5K model to 10K catalog), 0.422 (10K model to 15K catalog), and 0.444 (15K model to 20K catalog), respectively, thanks to the I-LLMRec’s cold-start item recommendation ability (See Appendix E.5). These results suggest that I-LLMRec can operate efficiently without frequent retraining, significantly reducing both computational complexity and training costs.
>
> Regarding system performance, I-LLMRec achieves a throughput of 20 users per second, including the indexing of visual features$^1$, across all catalog sizes. This assumes the use of a fast retrieval model to first retrieve relevant items from the larger pool, with I-LLMRec performing re-ranking within the top 100 candidates. The GPU memory requirements are as follows: 6.7GB to load the model, and 21GB of GPU memory during inference with a batch size of 16. In terms of storage, the model weights require 14.3GB, and the visual feature cache requires 3.4KB per item. Thus, for every 5K increase in the catalog size, an additional 17MB of storage is needed. However, as the item catalog grows, the I-LLMRec model weights replace the previous model weights, meaning that no additional storage is required.
>
> For cost per request, since LLM-based recommendation models are primarily used for re-ranking [1,2], the processing cost for a user remains constant regardless of catalog size, as indexing the visual features is negligible with a complexity of O(1). The key scaling bottleneck is the periodic retraining necessary to maintain performance as the catalog expands. Using a single NVIDIA GeForce A6000 48GB GPU, an additional 3 hours of training time is required for every 5K increase in catalog size. However, we anticipate that utilizing multiple GPUs for offline training and fine-tuning models trained on previous catalogs will substantially reduce training time, compared to the current approach, which retrains the model from scratch with each catalog size increase.
>
> To improve the quality of the paper and offer insight into the scalability of the item catalog, we have included it in Appendix E.13.
>
> [1] Large Language Models meet Collaborative Filtering: An Efficient All-round LLM-based Recommender System. Kim et al., KDD’24
>
> [2] LLaRA: Large Language-Recommendation Assistant. Liao et al., SIGIR’24
>
> 1. To facilitate the real-world recommendations, we pre-compute and cache the visual features, preventing the need for repeated computation of visual features and reducing the computational burden.
>
> ---

---

> ### Author Response · Authors · 2025-11-21
> **(2/3) Author Rebuttal**
>
> ### **W2: Relying on Images while discarding available text is counterintuitive**.
> We’d like to clarify that our work does not advocate discarding textual information or relying solely on images. The goal of I-LLMRec is to demonstrate that item images can serve as an efficient and effective representation for capturing user preferences in LLM-based recommenders—while textual descriptions remain available and are used as complementary signals rather than being removed. Specifically, we provide following analyses:
>
> * Appendix E.6 examines the missing-image scenario by substituting textual descriptions for absent images, demonstrating the practical applicability of I-LLMRec.
>
> * Appendix E.7 shows that I-LLMRec remains robust by leveraging textual descriptions when image quality is poor.
>
> Lastly, to examine performance when visual cues are intrinsically weak, we conduct experiments on the Goodbooks dataset. Even in this text-centric domain (i.e., book), I-LLMRec outperforms TALLRec, indicating that our approach generalizes beyond visually rich item categories. We also note that the description-based model TRSR performs competitively in such domains, and we discuss this comparison in Section E.4.
>
> Overall, these analyses illustrate that I-LLMRec does not discard or undervalue text. Instead, it remains effective across missing-image cases, poor-image-quality scenarios, and highly text-centric domains.
>
> | Method | GoodBooks (HR@5 / NDCG@5) |
> |--------|----------------------------|
> | SASRec | 0.4475 / 0.2956            |
> | TallRec | 0.4752 / 0.3266           |
> | I-LLMRec   | 0.4870 / 0.3372            |
>
> ---
> ### **W3: Comparisons lack strong compressed-text**.
> We’d like to clarify that our study already includes a compressed-text baseline (TRSR), which performs natural-language summarization to reduce the token budget of item representation. Nevertheless, to further strengthen our evaluation, we additionally compare against  LC-Rec [1], a state-of-the-art description-compression method that employs a vector quantization compact encoder to tokenize each item description into 4 discrete latent tokens. Despite this strong learned compression, I-LLMRec still achieves superior performance, as shown in table below. This indicates that item images with one visual token provide a more informative and efficient item representation for an LLM.
>
> Regarding fairness in trainable parameters, training wall-clock time, and inference budget, I-LLMRec is already more efficient than the compressed-text baselines. Our method introduces only 0.2% additional trainable parameters for a lightweight visual adapter and projection heads (F*_u and F*_i in the RERI module), and encodes each item with a single image token, greatly reducing computation during both training and inference. On the other hand, LC-Rec requires 6.3% additional trainable parameters for LoRA fine-tuning and an external compression model, and still relies on 4 tokens per item, increasing computational overhead.
>
> To further ensure fairness, we also evaluate a LoRA-fine-tuned version of I-LLMRec using 4.8% trainable parameters. Even under this matched setting, I-LLMRec further  outperforms LC-Rec, reinforcing that image-based representations remain more effective.
>
> Overall, these results demonstrate that item images provide a more efficient and effective item representation than text-based compressed pathways, thanks to the rich semantic information of item images.
>
> | Method                | Sport/Art (HR@5) | Trainable Parameters | Token Usage per Item |
> |-----------------------|-------------------------------|------------------------|------------------------|
> | LC-Rec (w/ LoRA)      | 15.86 / 20.41                 | 6.3%                   | 4                      |
> | I-LLMRec (w/o LoRA)   | 16.28 / 21.80                 | 0.2%                   | 1                      |
> | I-LLMRec (w/ LoRA)    | **16.99 / 23.58**             | 4.8%                   | 1                      |

---

> ### Author Response · Authors · 2025-11-21
> **(3/3) Author Rebuttal**
>
> ### **W4: Methodological fairness is unclear given image-side tuning and a frozen LLM**
> To address the reviewer’s concern about potential unfairness arising from tuning only the image adapter while keeping the LLM frozen, we conducted additional experiments where both modalities are given comparable adaptation capacity. Specifically, on the Amazon Sport and Art datasets, we applied LoRA-based finetuning to the LLM while simultaneously training the image adapter, using both image and textual descriptions as inputs (I-LLMRec+D). As shown in the table below, the resulting performance remains competitive, even when the LLM benefits from textual finetuning and the visual pathway benefits from the image adapter. These results suggest that allowing adaptation on the visual side while keeping the LLM frozen does not disadvantage textual baselines and thus does not compromise methodological fairness.
>
> | Method                 | Sport (Hit@5 / NDCG@5) | Art (Hit@5 / NDCG@5) |
> |------------------------|--------------------------|-------------------------|
> | I-LLMRec+D             | 0.4554 / 0.3637          | 0.5902 / 0.4796         |
> | I-LLMRec+D w/ LoRA     | 0.4552 / 0.3680          | 0.5832 / 0.4800         |

---

> > ### Comment · Reviewer_1MNz · 2025-11-25
> >
> > Thank you for the authors’ response; I have increased my score.

---

> ### Author Response · Authors · 2025-11-27
>
> We thanks the reviewers for raising the score to 8. We sincerely appreciate your valuable feedback, which has helped us further improve the paper, as well as the time you've dedicated to this process. We are glad to hear that our responses have met your expectations.

---

### Official Review · Reviewer_3dZW · 2025-10-31

**Soundness:** 2
**Presentation:** 3
**Contribution:** 3
**Rating:** 4
**Confidence:** 4

**Summary:**

The paper proposes I-LLMRec, an LLM-based recommender that replaces lengthy item descriptions with item images to represent a user’s interaction history. It introduces (1) RISA (Recommendation-oriented Image–LLM Semantic Alignment), which learns an adaptor to map visual features into the LLM space using prompt-based supervision on next-item properties (brand/category/title/description), and (2) RERI (Retrieval-based Recommendation via Image features), which retrieves items from the corpus with user representations derived from an LLM token [REC], projected to a shared space and compared via dot products. Experiments across four Amazon categories (Sports, Grocery, Art, and Phone) demonstrate that image-based representation significantly outperforms attribute- and description-based LLM baselines, achieving about 2.93× faster inference while delivering roughly 22% performance gains over attribute-based representation. Additionally, this approach shows enhanced robustness under tight context budgets and is more resilient to noisy descriptions.

**Strengths:**

1. **Significant and Practical Problem:** The paper addresses a critical and practical problem in LLM-based recommender systems: the trade-off between efficiency (low token usage) and effectiveness (rich semantic representation). It clearly diagnoses why attribute-based methods are efficient but semantically poor, while description-based methods are rich but computationally expensive.

2. **Excellent Efficiency and Scalability:** The core engineering contribution is demonstrating that using images as a prompt representation is highly efficient. Figure 3 provides strong evidence that the inference time of `I-LLMRec (I)` (red line) remains low and stable as the user sequence length (`|S_u|`) increases, whereas text-based methods like `TRSR (D)` (black line) and `I-LLMRec+D` (gray line) scale poorly due to their reliance on lengthy descriptions.

3. **Robustness to Context Window Limits:** The analysis in Figure 4 is a key strength. It shows that `I-LLMRec`'s performance is stable even with a severely constrained context window (e.g., 256 tokens). In contrast, description-based models suffer a sharp performance decline as the context window shrinks, highlighting `I-LLMRec`'s practical value in token-limited environments.

4. **Novel Alignment Methodology:** The proposed Recommendation-oriented Image–LLM Semantic Alignment (RISA) module is a clever contribution. Instead of relying on generic image–caption alignment, it trains the adaptor (M) to align visual features with recommendation-specific properties (e.g., brand, category, title, description). The ablation study (Table 2, row (a) vs. (b)) confirms that this task-specific alignment is crucial, leading to significant performance gains.

5. **Clear Analysis of Text Robustness:** Figure 5 provides a clear empirical argument for the robustness of images over text. It demonstrates that text-based approaches are vulnerable to either information loss from summarization (Fig. 5a) or performance degradation from noise in full descriptions (Fig. 5b).

**Weaknesses:**

1. **Title and Premise are Overclaimed:** The paper's title, "Image is All You Need", and its central premise are fundamentally contradicted by its own best-performing model. The SOTA results reported in Table 1 for "I-LLMRec (I)" are, according to the ablation study (Table 2, row (e)), achieved by a model that fuses Image + CF + Text features. This model explicitly requires textual features ($r_{u,i}^{lext}$) in its RERI retrieval module to achieve top performance. The paper does not replace text; it moves text from the (inefficient) LLM prompt to the (efficient) retrieval head. This is a major overclaim.
2. **Incomplete Efficiency Costing:** The efficiency gains reported in Figure 3 are potentially misleading. The paper does not clarify whether the inference time includes the (potentially significant) computational cost of extracting visual features using the SigLIP encoder. If these features are assumed to be pre-computed, the reported wall-clock times do not reflect the full cost of a real-time recommendation.
3. **Weak Justification:** The entire motivation rests on the "significant information overlap" between images and text 。The primary evidence is a CLIP cosine similarity of ~0.31 (Figure 1c) 。This value is not self-evidently "significant" or "surprisingly high." It is only marginally higher than COCO positive pairs (0.26)  and is very far from a strong correlation. This justification is weak and feels like an overstatement to fit the narrative.
4. **Flawed Argument on Textual Information:** The paper claims in Section 4.2 that text-specific information "has surprisingly little impact on recommendations" 。The evidence cited is that I-LLMRec+D (adding text descriptions to the prompt) did not improve performance。This is a flawed argument. This experiment only proves that adding text to the prompt is redundant (which Figure 3 confirms is also inefficient)，not that text information itself is useless. In fact, the paper's own ablation (Table 2, row (d) vs. (e)) directly disproves this claim, showing that adding "Text" features to "Image + CF" features improves performance (e.g., Sport Hit@5 from 0.4491 to 0.4570; Art Hit@5 from 0.5795 to 0.5883) 。This confirms that text is impactful and necessary for the model's best performance.
5. **Limited External Validity:** All datasets used are from the retail and fashion domains. These are visually-driven domains where this method is predisposed to succeed. The paper provides no evidence of generalization to visually-weak domains (e.g., books, news, services).

**Questions:**

1.  Your SOTA results in Table 1 (e.g., 0.4570 Hit@5 on Sport) appear to match row (e) of your ablation study in Table 2, which is the **"Image + CF + Text**" variant. How do you justify the title "Image is All You Need" when your best-performing model explicitly requires **text features** in its retrieval module? Given that your actual contribution seems to be moving text features from the (inefficient) LLM prompt to the (efficient) retrieval head, would a more accurate title be "Images as an Efficient Prompt Representation for Multimodal Recommender Systems"?

2. In Section 4.2, you claim text-specific information has "little impact," using the `I-LLMRec+D` experiment as evidence. However, your own ablation study (Table 2, row (d) vs (e)) shows that adding "Text" features to the retrieval module *improves* performance (e.g., Sport Hit@5 from 0.4491 to 0.4570) . How do you reconcile this direct contradiction?

3.  Does the `I-LLMRec+D` experiment (text in prompt) not simply prove that text in the *prompt* is redundant *if* text is already being used in the *retrieval head*, rather than proving that text information *itself* is useless?

4.  Do the inference times reported in Figure 3 include the wall-clock time for **visual feature extraction** (i.e., the SigLIP encoder pass)?

5.  If the visual features were pre-computed, what is the actual latency of this feature extraction step per item, and how does this cost compare to the LLM processing time (which you optimized)? In a real-time system, wouldn't this encoder cost be a significant bottleneck?

6.  You justify your entire premise on a CLIP cosine similarity of ~0.31, calling it "significant overlap". Given that a 0.31 correlation is generally considered weak, why do you believe this is strong enough evidence to claim that images can semantically *replace* detailed text descriptions?

7.  All your experiments were conducted on retail and fashion datasets (Sports, Art, Clothing, etc.) where items are inherently visual. How do you expect I-LLMRec to perform in **visually-weak domains**, such as recommending books (where the cover art has low semantic value), news articles, or financial services?

---

> ### Author Response · Authors · 2025-11-21
> **(1/2) Author Rebuttal**
>
> ### **W1(Q1): Titles are Overclaimed**.
>
> We agree that the title “Image is All You Need” may overstate our claim. Our original intention was to emphasize a shift in perspective: while prior works rely on natural language prompts to model user preferences for LLMs, our focus is to show that images can serve as more efficient and semantically rich prompts. Regarding the retrieval head, our aim is not to complement user preference with text features; rather, we use textual features to illustrate that the retrieval module can flexibly incorporate any modality that provides item-level embeddings beyond images. Text is simply a practical example due to its availability; the same framework could integrate other modalities (e.g., acoustic features) without conceptual changes.
>
> Therefore, we kindly ask reviewers to view our main contribution as improving LLM-based user preference modeling via image features, with the textual features in the retrieval head serving only to demonstrate extensibility rather than contradicting our premise. Still, we acknowledge that the current title may be misleading, and, following the reviewer’s suggestion, we are willing to update it to: “Images as an Efficient and Effective Prompt Representation for LLM-based Recommender Systems.”
>
> ---
> ### **W2(Q4,5): Incomplete Efficiency Costing**.
>
> We thank the reviewer for raising this concern and would like to clarify how we handle visual feature extraction in our efficiency evaluation. Following prior multimodal recommendation studies [1–3], we pre-compute and cache all item visual features before both training and inference. Thus, the wall-clock times in Figure 3 exclude SigLIP extraction cost. However, we argue that this setup aligns with real-world recommendation scenarios: once an item’s visual features are extracted offline, the same representation can be reused for every user who interacts with that item, during both training and inference. Prior work [1–2] adopts the same convention when evaluating efficiency. Furthermore, it is important to note that caching these features is inexpensive—only 3.4KB per item, making the memory requirement extremely lightweight.
>
> To further address the reviewer’s concern, we also measure inference time when including visual feature extraction. On the Sports dataset, TRSR (description-based) requires about 47 minutes for recommendation inference, while I-LLMRec requires 15 minutes for recommendation inference plus 5 minutes for feature extraction (~20 minutes total). Even under this setting, I-LLMRec is 2.4 times more efficient.
>
> *Processing Time (Q5)*: Extracting visual features takes 0.03 seconds per item, which is 58% faster than an LLM forward pass. As noted earlier regarding pre-computation and caching—where each item can be processed only once and then reused—this cost does not present a practical bottleneck.
>
> To avoid confusion regarding the efficiency analysis, we have included information about the precomputation of visual features in Lines 385-387.
>
> [1] Improving Multi-modal Recommender Systems by Denoising and Aligning Multi-modal Content and User Feedback. Xv et al., KDD’24.
>
> [2] COHESION: Composite Graph Convolutional Network with Dual-Stage Fusion for Multimodal Recommendation. Xu et al., SIGIR’25.
>
> [3] GUME: Graphs and User Modalities Enhancement for Long-Tail Multimodal Recommendation. Lin et al., CIKM’24.
>
> ---
> ### **W3(Q6): Weak Justification**.
>
> We kindly ask the reviewer to interpret the CLIP cosine similarity of 0.31 from a relative rather than absolute perspective. Specifically, **in the well-curated COCO image–caption dataset**, positive image–caption pairs have a cosine similarity of 0.26, while negative captions yield 0.07 (Line 97), creating a gap of 0.19. Under this comparison, our similarity score of 0.31 is 0.05 higher than well-curated COCO’s positive-pair value—corresponding to a relative increase of approximately 26% when scaled by the positive–negative gap.
>
> It is important to note that this 26% increase appears even though COCO contains carefully aligned, human-curated image–caption pairs. This gap suggests that a similarity of 0.31 shows a high degree of semantic overlap.

---

> ### Author Response · Authors · 2025-11-21
> **(2/2) Author Rebuttal**
>
> ### **W4(Q2,Q3): Flawed Argument on Textual Information**.
>
> We’d like to clarify the difference between the statement in Lines 361–363—“text-specific information has little impact on recommendation”—and the performance gap between rows (d) and (e) in Table 2. The statement in Lines 361–363 concerns the *LLM-based user preference understanding stage*, where additional textual information provides only limited benefit beyond what the model already captures from images. In contrast, the improvement from row (d) to row (e) reflects the fact that *during the retrieval stage*, text-based item embeddings can indeed enhance recommendation accuracy. Thus, our point is not that text is unnecessary, but that its function differs between the user preference modeling stage and the retrieval stage.
>
> Furthermore, we do not argue that textual information is broadly unimportant. Its usefulness is evident from several points: 1) TRSR (with descriptions) outperforms TALLRec (w/o descriptions), 2) text information can be complementary when images are missing (Appendix E.6), and 3) row (h) improves upon row (e) in Table 2.
>
> Our intention in Lines 361–363 was simply to highlight that images alone can effectively capture user preference. However, as the reviewer pointed out, the phrasing could be misleading. To avoid confusion, we have removed this sentence from the manuscript, shown in Line 370-374.
>
> ---
> ### **W5(Q7): Limited External Validity**
>
> To explore the behavior of I-LLMRec on visually weak domains, we conducted additional experiments on the Goodbooks dataset with multiple baselines (SASRec, TALLRec, and TRSR). As shown in the table, the description-based model, TRSR, outperforms I-LLMRec on this dataset. This aligns with intuition: in visually weak domains, visual features carry limited information for capturing user preferences, and textual descriptions naturally become more informative.
>
> However, we highlight two important points. First, while long textual descriptions can help in fully text-driven domains like Goodbooks, relying on them is often impractical due to significant computational cost. For example, TRSR incurs 2.8× longer inference time and 3.7× longer training time than I-LLMRec on Goodbooks. Furthermore, although we can leverage descriptions in such domains due to their low image–description cosine similarity (0.16 compared to over 0.29 in other datasets), these domains are limited in scope. Second, across the eight datasets used in the paper—including Sport and Phone, which can also be considered visually weak—we consistently observe the high degree of semantic overlap between item images and descriptions, and I-LLMRec performs effectively, demonstrating that I-LLMRec generalizes beyond purely visually rich domains (e.g., Fashion). In summary, the efficiency and effectiveness of I-LLMRec across diverse domains make it broadly more suitable for real-world recommendation scenarios.
>
> Lastly, in response to reviewer comments regarding Weakness 1 and 5, we acknowledge that the original title, “Image is All You Need,” was overstated. We have therefore revised it to “Images as an Efficient and Effective Prompt Representation for LLM-based Recommender Systems.”
>
> We appreciate the reviewer’s insightful feedback on evaluating visually weak domains. To illustrate the broader behavior of I-LLMRec and strengthen the paper, we have added the corresponding analysis in Appendix E.4.
>
> | Method   | (HR@5 / NDCG@5) |
> |----------|----------------------------|
> | SASRec   | 0.4475 / 0.2956            |
> | TallRec  | 0.4752 / 0.3266            |
> | TRSR     | **0.5100** / **0.3551**            |
> | I-LLMRec | 0.4870 / 0.3372            |

---

### Official Review · Reviewer_xUwd · 2025-11-02

**Soundness:** 2
**Presentation:** 2
**Contribution:** 2
**Rating:** 2
**Confidence:** 4

**Summary:**

This paper studies the trade-off between efficiency and effectiveness in LLM-based recommendation. Existing methods rely on either attribute-based or description-based item representations, which are limited by low semantics or high token cost. The authors observe strong information overlap between images and descriptions and propose I-LLMRec, which uses images as compact item representations. The model includes two modules: RISA for image–language alignment and RERI for retrieval-based recommendation. Experiments on multiple real-world datasets show that I-LLMRec improves both efficiency and accuracy.

**Strengths:**

1.	The motivation is clear, and the trade-off is well analyzed.
2.	The proposed method is overall reasonable and effective.
3.	Experiments are extensive and convincing.
4.	The paper is well organized and easy to follow.

**Weaknesses:**

1.	The paper lacks discussion or comparison with other generative textual item identifier such as IDGenRec [1] and semantic ID methods such as LC-Rec [2] or SETRec [3].
2.	The experiments are limited to Amazon datasets. The generalization to other domains such as micro-video recommendation (e.g., MicroLens) is necessary to strengthen the paper.
3.	Lack necessary ablation study on the image data, which is a key component of the proposed method.

[1] Tan et al., IDGenRec: LLM-RecSys Alignment with Textual ID Learning. SIGIR’24
[2] Zheng et al., Adapting Large Language Models by Integrating Collaborative Semantics for Recommendation. ICDE’24.
[3] Lin et al., Order-agnostic Identifier for Large Language Model-based Generative Recommendation. SIGIR’25.

**Questions:**

1.	How does I-LLMRec perform on non-e-commerce datasets, such as MicroLens?
2.	Can the authors compare token efficiency with semantic ID approaches such as LC-Rec or SETRec?
3.	How significant is the contribution of the image modality? The ablation study should include the model using CF and text only, without image features.

---

> ### Author Response · Authors · 2025-11-21
> **(1/2) Author Rebuttal**
>
> ### **W1(Q2) Comparison and Discussion with Generative Textual Item Identifier and Semantic ID methods**
>
> We’d like to clarify the differences between the Semantic ID approach and our approach. While both approaches share the goal of efficiently representing items within the LLM’s capacity, there is a fundamental difference in how they convey the semantics of item descriptions to the LLM. The Semantic ID approach addresses the challenges of randomly initialized item tokens—which inherently fail to capture semantic relations between items [1]—by training a vector-quantized model that maps item descriptions to discrete codes. These descriptions serve as semantic intermediaries, aiding in the capture of semantic relationships between items (i.e., if two items share similar descriptions, they typically have the same codes), and allowing the LLM to ***implicitly*** capture the semantics of the descriptions. On the other hand, our approach ***explicitly*** encodes the semantics of each item’s description using images, which provide a richer and more direct representation for the LLM.
>
> Regarding token efficiency, Semantic ID approaches [2,3] generally require four tokens per item, whereas our approach uses a single image token that captures the semantics of the description. Importantly, Semantic ID methods essentially require multiple discrete tokens to express cross-item semantic relationships, while our approach conveys the semantics of the description with just one image token.
>
> Beyond token efficiency, we also compare performance with a state-of-the-art semantic ID method, LC-Rec, under a fair evaluation setting using the same LLM backbone on the Sport and Art datasets with full item candidates. I-LLMRec achieves Hit@5 scores of 21.80 (Sport) and 16.28 (Art), compared to 20.41 and 15.86 for LC-Rec, respectively. These results indicate that I-LLMRec not only offers more efficient item representation but also more effective in representing item semantics to the LLMs.
>
> We are grateful to the reviewer for allowing the exploration of the discussion on the semantic ID approach, and we have incorporated it into Section D.3. **In addition, we include a discussion of the textual item identifier approach in the same section**.
>
> [1] Understanding Generative Recommendation with Semantic IDs from a Model-scaling View. Liu et al., Arxiv’25.
> [2] Zheng et al., Adapting Large Language Models by Integrating Collaborative Semantics for Recommendation. ICDE’24.
> [3] Lin et al., Order-agnostic Identifier for Large Language Model-based Generative Recommendation. SIGIR’25.
>
> ---
> ### **W2(Q1): Experiments on Non-Ecommerce Dataset (e.g., MicroLens)**
>
> We appreciate the reviewer’s suggestion regarding the need to evaluate our method beyond the Amazon datasets. To address this, we have already conducted additional experiments using a dataset from a different domain (H&M), as shown in Appendix E.4. This experiment demonstrated the effectiveness of our method and suggested I-LLMRec’s generalizability across different dataset platforms.
>
> Furthermore, in response to the reviewer’s suggestion, we extended our evaluation to the micro-video recommendation domain by using the MicroLens dataset. For this, we extracted visual features by averaging the visual features of sampled frames provided by the MicroLens dataset. The results, as shown in the table below, show that TALLRec performs similarly to a traditional collaborative filtering model (SASRec), highlighting that user preferences based solely on titles are insufficient. On the other hand, I-LLMRec significantly outperforms TALLRec, demonstrating the effectiveness of incorporating visual features and further confirming the generalization of I-LLMRec across different domains.
>
> To further enhance the quality of the paper, we have included this experiment in Section E.4.
>
> | Model    | HR@5 / NDCG@5      |
> |----------|---------------------|
> | SASRec   | 0.4567 / 0.3413     |
> | TALLRec  | 0.4459 / 0.3237     |
> | I-LLMRec | **0.4854** / **0.3520**     |

---

> ### Author Response · Authors · 2025-11-21
> **(2/2) Author Rebuttal**
>
> ### **W3(Q3): Lack of Ablation Study on the Image Modality**
>
> To address the reviewer’s concern regarding the lack of an ablation study on the image modality, we conducted additional experiments that include (1) models using a single modality (Image, CF, or Text) and (2) models using all pairwise combinations of two modalities. These results are shown below and have been added to Table 2 in the revised main paper.
>
> From the single-modality results, we find that the image-only variant (row (b)) outperforms both the CF-only (row (c)) and Text-only (row (d)) variants, indicating that the image modality provides a key contribution on its own. For the two-modality settings, all combinations perform comparably, while the model using all three modalities (row (h)) achieves the best overall performance.
>
> We would like to clarify that our goal in incorporating CF and Text features is to demonstrate that our RERI module effectively extends to multiple modalities (refer to “Extension to multiple feature types” in Section 3.3)—not to claim that the image modality remains dominant when combined with others.
>
> We appreciate the reviewer’s valuable feedback regarding the ablation studies and have revised the ablation study section in Section 4.4 accordingly.
>
> | Row | RISA | RERI (Image) | RERI (CF) | RERI (Text) | Sport Hit@5 | Sport NDCG@5 | Art Hit@5 | Art NDCG@5 |
> |-----|------|-------------|----------|-------------|--------------|---------------|------------|--------------|
> | (a) | ✓    |             |         |             | 0.3953       | 0.3043        | 0.5040     | 0.3915       |
> | (b) | ✓    | ✓           |          |            | 0.4316       | 0.3403        | 0.5564     | 0.4447       |
> | (c) | ✓    |             | ✓        |            | 0.4075       | 0.3256        | 0.5502     | 0.4517       |
> | (d) | ✓    |           |          |    ✓          | 0.4178       | 0.3309        | 0.5522     | 0.4420       |
> | (e) | ✓    |✓             | ✓        |            | 0.4491       | 0.3630        | 0.5795     | 0.4769       |
> | (f) | ✓    |            | ✓        |    ✓         | 0.4442       | 0.3594        | 0.5786     | 0.4758       |
> | (g) | ✓    | ✓           |         |    ✓         | 0.4494       | 0.3602        | 0.5790     | 0.4759       |
> | (h) | ✓    | ✓           | ✓        | ✓           | **0.4570**   | **0.3711**    | **0.5883** | **0.4839**   |

---

### Author Response · Authors · 2025-11-29

Dear AC,

Thank you for taking on the difficult task of reviewing many papers on such short notice. To help minimize your workload, we provide a brief summary of the rebuttal process and the reviewers’ comments for our submission.

## **Rebuttal Score Changes**

**R1**: Reject (2) → NA
**R2**: Marginal Below (4) → NA
**R3**: Marginal Above (6) → Accept (8)
**R4**: Accept (8) → Strong Accept (10)
Only R3 and R4 participated in the rebuttal discussion, and both acknowledged the clarifications provided and increased their scores accordingly. Evidence of these rating updates can be found at [[R3 Response]](https://openreview.net/forum?id=vizM7B7vuW&noteId=tr1dPtDDM5) and [[R4 Response]](https://openreview.net/forum?id=vizM7B7vuW&noteId=0ul1MhpP5m).

## **Summary of R1 and R2 Feedback and Our Responses**.

Since R1 and R2 did not participate in the discussion, we summarize their concerns and our responses for your convenience.

### **R1 Feedback & Our Response**.
* Regarding the *Discussion and comparison with generative textual item identifiers and semantic ID approaches*, we incorporated the detailed discussion and efficiency/performance comparison with these approaches, demonstrating that our approach is more efficient and effective than these approaches. [[Response link]](https://openreview.net/forum?id=vizM7B7vuW&noteId=AvsFeo4CB8)

* Regarding the *Need for experiments on non-commerce datasets (e.g., MicroLens)*, we included results demonstrating effectiveness on MicroLens dataset. [[Response link]](https://openreview.net/forum?id=vizM7B7vuW&noteId=AvsFeo4CB8)

* Regarding the *Additional ablations on the image modality*, we provided detailed ablation studies of image modality.  [[Response link]](https://openreview.net/forum?id=vizM7B7vuW&noteId=zzB49s4Yca)

### **R2 Feedback & Our Response**
* Regarding the *Potential overclaim in the title*, we clarified the intended meaning of the title. [[Response link]](https://openreview.net/forum?id=vizM7B7vuW&noteId=WasFw7NW8D)

* Regarding the *Need for a complete explanation of the efficiency cost of handling image features*, we provided a complete clarification of the image feature cost, along with comparisons that include the cost of processing image features. [[Response link]](https://openreview.net/forum?id=vizM7B7vuW&noteId=WasFw7NW8D)

* Regarding the *Further justification for image-description information overlap*, we justify information overlap from a relative rather than absolute perspective. [[Response link]](https://openreview.net/forum?id=vizM7B7vuW&noteId=WasFw7NW8D)

* Regarding the *Flawed Argument on Textual Information*, we clarified that textual information is complementary, not useless as the reviewer seemed to infer. [[Response link]](https://openreview.net/forum?id=vizM7B7vuW&noteId=QnqAv28yDw)

* Regarding the *Request for experiments on visually weak domains*, we included results on the Goodbooks dataset. [[Response link]](https://openreview.net/forum?id=vizM7B7vuW&noteId=QnqAv28yDw)

Although R1 and R2 did not engage in this rebuttal, we believe these responses clearly addressed their concerns, and we kindly ask you to consider them.

Thank you very much for your time and effort.

---

### Meta-Review · Area_Chair_oiTn · 2026-01-05

**Summary:**

The reviewers raised several substantive concerns centering on (i) overclaimed novelty in positioning images as exclusive inputs without adequate distinction from semantic-ID methods (e.g., LC-Rec), (ii) incomplete empirical validation including missing efficiency analysis (e.g., image encoding overhead), limited domain coverage beyond e-commerce (e.g., visually weak domains like books), and insufficient ablation studies isolating image modality contributions, (iii) methodological fairness in comparisons (e.g., baselines lacking image-enhanced variants), and (iv) scalability under industrial constraints (e.g., cold-start scenarios and growing catalog sizes).

**Reviewer Concerns:**

The rebuttal comprehensively addressed core issues:
1. Clarified that images serve as efficient prompts rather than exclusive inputs, revised the title to reflect text-image complementarity, and added fallback strategies for missing images.
2. Provided granular efficiency analysis, expanded validation to visually weak domains (Goodbooks) and non-commerce data (MicroLens), and conducted rigorous ablation studies.
3. Ensured methodological fairness by comparing against image-enhanced TALLRec/LC-Rec variants and LoRA-finetuned semantic-ID baselines under matched parameter budgets.

**Reviewer Scores:**

Reviewer xUwd was confident in a reject recommendation and would be unlikely to change their score. Reviewer 3dZW rated the paper marginally below threshold but appreciated its novelty; their score would likely increase to marginally above threshold. Reviewer 1MNz initially rated the paper marginally above threshold and, following the rebuttal and additional experiments, explicitly stated they would raise their score. Reviewer 11jm initially rated the paper as “accept – good paper (poster)” and, after reviewing the rebuttal and new experimental results, also explicitly indicated they would increase their score.

---

### Decision · Program_Chairs · 2026-01-26

Accept (Poster)